# OpenReview forum: "ReGenesis: LLMs can Grow into Reasoning Generalists via Self-Improvement"
_ICLR.cc/2025/Conference — ICLR 2025 Oral_

### Official Review · Reviewer_TzKs · 2024-10-23

**Soundness:** 3
**Presentation:** 4
**Contribution:** 3
**Rating:** 8
**Confidence:** 4

**Summary:**

This paper proposes ReGenesis which applies task-agnostic reasoning guidance to facilitate reasoning path synthesis, achieving great performance in a self-improvement manner. Experiments on in-domain and out-of-domain tasks show the strong generalization ability of the proposed data-synthesizing and training framework. Further analysis indicates the importance of each component in ReGenesis and demonstrates the robustness across different base models.

**Strengths:**

- The framework of ReGenesis is clearly shown, especially in Figure 1.
- The generalization ability of the proposed method is impressive, even in a self-improvement manner.
- Empirical studies in this paper are comprehensive.
- The paper is well-written

**Weaknesses:**

1. Missing some baselines: How will ReGenesis perform when compared with few-shot CoT prompting (w/o FT) on in-domain and out-of-domain tasks, and how will it perform compared with GPT-3.5/4/4o? Also, if Mistral-7B is fine-tuned on gold trajectories, which means removing the self-improvement setting, how large will the performance gap be between this and ReGenesis?
2. Why is A+S not included in Table 4?

3. StrategyQA and OpenbookQA are not challenging for LLMs nowadays. What would happen if training is done on StrategyQA and testing is performed on CommonsenseQA, a more challenging out-of-domain dataset? Furthermore, ASDIV, SVAMP, and AQUA are quite simple for GSM8K. If training is conducted on these datasets and testing is done on GSM8K, what would happen?

**Questions:**

1. It appears that the generalization of ReGenesis mainly stems from task-agnostic guidance. However, during actual model training, only specific trajectories are provided, without including this guidance. So, where does the model's generalization come from? Fundamentally, task-agnostic guidance simply ensures that the model can synthesize more correct trajectories. These additional trajectories enhance the model's capabilities. If ReGenesis, STaR, and LMSI were trained using the same number of synthetic trajectories, would there be differences in their effectiveness?
2. Could the authors provide a detailed analysis and explanation as to why giving hints to models when generating reasoning paths might have adverse effects?

---

> ### Author Response · Authors · 2024-11-21
> **Summary of Response**
>
> We appreciate Reviewer TzKs's valuable feedback, which has guided us to conduct additional empirical analysis that strengthens our submission. In response, we present the following analysis:
>
> 1. **Why ReGenesis works**: We conduct empirical analysis to demonstrate why **ReGenesis** outperforms competing baselines, emphasizing that **data size** is not a contributing factor to our model's success.
> 2. **Additional Baseline: Few-shot CoT Prompting**: Following the reviewer's suggestion, we introduce an additional baseline: **few-shot CoT prompting** (w/o FT), using **GPT-4-generated CoT examples** for testing the **Mistral-7B-Instruct-v0.3** model.
> 3. **Fine-tuning on GSM8k’s Gold Trajectories**: We fine-tune the **Mistral-7B-Instruct-v0.3** model on **GSM8k’s Gold Trajectories** and show that it still underperforms compared to the **ReGenesis-trained model** in both **in-domain** and **out-of-domain** scenarios.
> 4. **Reasoning Path Hints & Adverse Effects**: We explain why providing **hints to LLMs** during **final reasoning path generation** has **adverse effects**, supported by relevant examples.
> 5. **Training-Testing Dataset Complexity**: We discuss the reviewer’s concern regarding our use of more complex training datasets and simpler test datasets, and we would like to clarify that this is not the case.
> 6. **Typo Correction**: We provide a correction for the typo ("A+S") noted by the reviewer.
>
> These revisions are intended to provide further clarification of our methods and findings, and we hope they adequately address all the points raised by the reviewer.

---

> ### Author Response · Authors · 2024-11-21
> **[1/5] Investigating Why ReGenesis Outperforms Competing Baselines**
>
> Reviewer: _“It appears that the generalization of ReGenesis mainly stems from task-agnostic guidance. However, during actual model training, only specific trajectories are provided, without including this guidance. So, where does the model's generalization come from?”_
>
> * Reviewer: _“More specifically,  task-agnostic guidance simply ensures that the model can synthesize more correct trajectories.” Is the data size difference the main reason for ReGenesis’s better generalization?_
>
>   **Our response**:
>
>   We thank the reviewer for this question since it points out that the size of the synthetic data after filtering is also a variable that could significantly affect model performance. We had this variable controlled in all the experiments of the current submission and we should have presented it in the experiment section. Although ReGenesis leads to slightly more correct paths,  we always down sample ReGenesis’s data to make sure its data size is the same as the rest baselines. The table below lists the training data size that ReGenesis and star use for all training data sets:
>
>
> |      | GSM8K  | NumGLUE | ARC_c | ReClor | StrategyQA |
> |------|--------|---------|-------|--------|------------|
> | Size | 31,994 | 85,078  | 4,789 | 18,086 | 4,079      |
>
> * Reviewer: _”If so, then where does ReGenesis’s generalization come from?”_
>
>   **Our response**:
>
>   **Although ReGenesis uses the final detailed and task-specific reasoning paths as training data, the diverse task-agnostic reasoning strategies that Regenesis adopts to generate such final task-specific paths, are effectively preserved in the final task-specific paths such that it is easy for LLMs, such as mistral-7B-instruct, to detect them and learn them from the final task-specific paths.** We run the following analysis to verify this hypothesis.
>
>
>   We randomly select 500 reasoning paths from ReGenesis and STaR’s generated data of _NumGLUE_ training data set. Note that the 500 reasoning paths of each method are all final task-specific reasoning paths. Then, we prompt _Mistral-7B-Instruct-v0.3_ to label each of the total 1000 reasoning paths the task-agnostic reasoning strategy behind the task-specific path. The goal of this experiment is to see whether from the task-specific path only, LLMs can already tell the task-agnostic strategy behind it and that ReGenesis’s task-specific paths have more diverse task-agnostic strategies. If so, it indicates that LLMs are likely to be able to capture the task-agnostic strategies behind the task-specific paths in the training data, even if they are trained only with  task-specific paths.
>
>
>   Specifically, after _Mistral-7B-Instruct-v0.3_ labeling the task-agnostic reasoning strategies, we task Mistral-7B-Instruct-v0.3 to cluster the task-agnostic reasoning strategies generated by itself and suggest cluster names for both ReGenesis and STaR. We present the results as follows. **Results show that it is quite apparent for Mistral-7B-Instruct-v0.3 that the final task-specific paths generated by ReGenesis, have more diverse task-agnostic reasoning strategies behind them, compared to STaR (More strategy clusters for ReGenesis and less for STaR).** Note that throughout this experiment, we don’t give Mistral-7B-Instruct-v0.3 any knowledge on how ReGenesis generates the final task-specific paths and what are the general reasoning strategies used by ReGenesis in the first two data generation steps. **On this condition, Mistral-7B-Instruct-v0.3 can still back-engineer and mine the diverse task-agnostic reasoning paths behind them, suggesting that diverse task-agnostic reasoning strategies used by ReGenesis are effectively preserved in the final task-specific reasoning paths, and that Mistral-7B-Instruct-v0.3 can detect them easily.**
>
>   We have also run the same analysis at OOD inference on _OpenBookQA_, on models fine-tuned with ReGenesis and STaR separately on NumGLUE dataset to show that models trained with ReGenesis, actually learned to adopt more diverse task-agnostic reasoning strategies at inference in OOD settings. We present the results below. **Results suggest that models trained with ReGenesis’s task specific reasoning paths, have learned the diverse task-agnostic reasoning strategies that ReGenesis has adopted for path generation and can apply them on OOD tasks.**
>
>
>   **_We show the clustering results on NumGLUE training data and OpenbookQA OOD inference data below separately._**

---

> > ### Author Response · Authors · 2024-11-21
> > **[2/5] Investigating Why ReGenesis Outperforms Competing Baselines**
> >
> > We show the clustering results of task-agnostic reasoning strategies mined from **STaR's** **NumGLUE training data**. (We only include examples for less straightforward cluster names due to the limit of the space.)
> >
> > **[STaR] Clusters with examples:**
> >
> > * Cluster 1: **Direct Calculation and Simplification**
> > * Cluster 2: **Algebraic and Formula-based Approaches**
> >     * **Question**: Claire decides to go to a nearby animal farm that has a total of 19 animals. He is standing in a queue near the entry gate, from the gate he could only see legs of the animals in the farm and he counted the total number of legs to be 56. He already knows that there are only ducks and horses in this farm. From this information find the number of ducks
> >     * **STaR Answer**: Let's say the number of ducks is D and the number of horses is H.\n\nSince each duck has 2 legs and each horse has 4 legs, the total number of legs can be represented by the equation:\n\n2D + 4H = 56\n\nSince there are 19 animals in total, we can set up another equation:\n\nD + H = 19\n\nWe can solve this system of equations by substituting D with 19 - H, which gives us:\n\n2(19 - H) + 4H = 56\n\nExpanding the equation and simplifying, we get:\n\n38 - 2H + 4H = 56\n2H = 18\nH = 9\n\nNow that we know the number of horses (H = 9), we can find the number of ducks (D) by substituting H into the first equation:\n\n2D + 4(9) = 56\n2D + 36 = 56\n2D = 20\nD = 10\n\nSo, there are 10 ducks in the farm.
> >     * **LLM Summarized Strategy**: How can I create equations that represent the problem and use them to find a solution?
> >
> > * Cluster 3: **Stoichiometry and Chemical Reactions**
> >
> >     * **Question**: How many moles of CH4 are required to react with 2 moles of Cl2 to form 2 moles of CH3Cl along with 2 moles of HCl
> >     * **STaR Answer**: The equation for the reaction is:\n\nCH4 (g) + Cl2 (g) \u2192 CH3Cl (g) + HCl (g)\n\nFrom the equation, we can see that:\n\n1 mole of CH4 reacts with 1 mole of Cl2 to form 1 mole of CH3Cl and 1 mole of HCl\n\nSince the question asks for 2 moles of CH3Cl and 2 moles of HCl, we can see that 2 moles of CH4 are required to react with 2 moles of Cl2 to form 2 moles of CH3Cl and 2 moles of HCl.\n\nTherefore, the answer is 2 moles of CH4.
> >     * **LLM Summarized Strategy**: Use direct application of stoichiometry: Identify the chemical reaction, balance it if necessary, and use the stoichiometric coefficients to directly calculate the required amount of reactants or products based on the given data.
> >
> > * Cluster 4: **Problem Decomposition and Step-by-Step Calculation**
> > * Cluster 5: **Logical Reasoning and Pattern Recognition**

---

> > > ### Author Response · Authors · 2024-11-21
> > > **[3/5] Investigating Why ReGenesis Outperforms Competing Baselines**
> > >
> > > We show the clustering results of task-agnostic reasoning strategies mined from **ReGenesis's** **NumGLUE training data** . (We only include examples for less straightforward cluster names due to the limit of the space.)
> > >
> > > **[ReGenesis] Clusters with examples**:
> > >
> > > * Cluster 1: **Step-by-Step Process**
> > > * Cluster 2: **Applying Logic and Formulas**
> > > * Cluster 3: **Reflective Thinking**
> > >     * **Question**: Joshua became headmaster of a school in the month of May. Joshua stayed headmaster for 2 months before retiring in the same year. In which month did Joshua retire?
> > >     * **ReGenesis Answer**:
> > > Step 1: Reflect on initial biases and assumptions
> > > I recognize that the common Western calendar year structure consists of 12 months, but I should be open to alternative structures and month lengths.
> > > Step 2: Reflect on past experiences and learning.
> > > I've encountered problems that required creative thinking and consideration of different scenarios and structures. I'm prepared to think creatively and consider alternative month lengths or school calendar systems.
> > > Step 3: Reframe the problem and ask specific and open-ended questions.
> > > We know that Joshua became headmaster in May ...
> > > Step 4: Analyze and evaluate the problem.
> > > Joshua was headmaster for 2 months, and we know he retired ...
> > > Step 5: Make a decision or solution.
> > > Based on the analysis, we can conclude that Joshua retired in July, considering the Western calendar structure, or July or August considering alternative month lengths and school calendar systems.
> > > Final Answer: \\boxed{July}
> > >
> > >     * **LLM Summarized Strategy**:  Reflective Thinking: This strategy emphasizes stepping back from the problem, engaging in introspection and self-reflection, examining personal biases and assumptions, and considering alternative interpretations or structures.
> > >
> > > * Cluster 4: **Sequential and Chronological Analysis**
> > > * Cluster 5: **Information Extraction**
> > > * Cluster 6: **Problem Decomposition**
> > > * Cluster 7: **Systematic Thinking**
> > >     * **Question**: How many moles of Sodium bicarbonate are required to react with 3 moles of Hydrochrolic acid to form 3 moles of Water, 3 moles of Carbon dioxide and 3 moles of Sodium chloride
> > >     * **ReGenesis Answer**: Step 1: Define the System. Reactants: 3 moles of Hydrochloric acid (HCl) and x moles of Sodium bicarbonate (NaHCO3). Products: 3 moles of Water (H2O), 3 moles of Carbon dioxide (CO2), and 3 moles of Sodium chloride (NaCl). Step 2: Identify the Reaction. Chemical reaction between HCl and NaHCO3 to form H2O, CO2, and NaCl. Reactants: 3 moles of HCl and x moles of NaHCO3. Products: 3 moles of H2O, 3 moles of CO2, and 3 moles of NaCl. Step 3: Analyze Feedback Loops and Interdependencies. The reaction is irreversible, meaning reactants are consumed and cannot be reversed. Formation of H2O and CO2 affects reaction rate and NaCl formation. Formation of NaCl affects reaction rate and H2O and CO2 formation. Step 4: Develop a Holistic Solution. Set up a balanced chemical equation:
> > > 3 HCl + NaHCO3 → 3 H2O + 3 CO2 + NaCl. Step 5: Determine the Required Amount of NaHCO3. 1 mole of NaHCO3 reacts with 1 mole of HCl to form 1 mole of each product. To react 3 moles of HCl, 3 moles of NaHCO3 are required. Conclusion: 3 moles of Sodium bicarbonate (NaHCO3) are required to react with 3 moles of Hydrochloric acid (HCl) to form 3 moles of Water, 3 moles of Carbon dioxide, and 3 moles of Sodium chloride.
> > > Therefore, the answer is: \\boxed{3 moles} of Sodium bicarbonate are required.
> > >
> > >     * **LLM Summarized Strategy**: Use systems thinking: Consider the problem as part of a larger system and understanding the interconnectedness of various elements. Focuses on identifying the underlying causes, feedback loops, and interdependencies that influence the problem, and developing holistic solutions that address the system as a whole.
> > >
> > > * Cluster 8: **Questioning Assumptions and Critical Thinking**
> > >     * **Question**: Piper counted wings of all commercial planes placed in an air exhibition. If total wings were 90, how many planes were there?
> > >     * **ReGenesis Answer**: I. Establish the known facts: Total number of wings = 90. Each plane has two wings. II. Question initial assumption: Is the problem asking us to find the total number of wings or the number of planes? III. Challenge and revise initial assumption: If each plane has two wings, then we can set up an equation: 2x = 90, where x is the number of planes. Divide both sides by 2 to solve for x: x = 90 ÷ 2 = 45. IV. Conclusion: Therefore, the correct answer is that there are \\boxed{45} planes in the air exhibition.
> > >   * **LLM Summarized Strategy**: Start by establishing known facts, questioning assumptions, and setting up a simple equation based on logical deduction. Simplify the problem by considering the relationship between the number of wings and planes, and solve for the unknown variable.
> > > * Cluster 9: **Mathematical and Analytical Calculation**
> > > * Cluster 10: **Stoichiometry and Chemical Problem Solving**

---

> > > > ### Author Response · Authors · 2024-11-21
> > > > **[4/5] Investigating Why ReGenesis Outperforms Competing Baselines**
> > > >
> > > > We show the clustering results of task-agnostic reasoning strategies mined from ***STaR's** **OpenbookQA OOD inference** data. (We only include examples for less straightforward cluster names due to the limit of the space.)
> > > >
> > > > **[STaR] Clusters:**
> > > > * Cluster 1 **Process of Elimination**: Focuses on systematically ruling out incorrect options to identify the correct answer.
> > > >     * **Question**: Where might a bunny live? (A) a thicket (B) atop palm trees (C) a sewer system (D) a deserted island
> > > >     * **STaR Answer**: A bunny would most likely live in a thicket, which is a dense growth of bushes or undergrowth, providing shelter and food. The other options are unlikely, as palm trees are tall and not typically found in habitats suitable for bunnies, sewer systems are not a suitable habitat for any animal, and a deserted island would be too isolated and lack the necessary food and shelter for a bunny. Therefore, the answer is (A).
> > > >     * **LLM Summarized Strategy**: Process of elimination: Assess each potential answer choice and eliminate those that are clearly incorrect or implausible. This strategy involves evaluating the likelihood or suitability of each option based on the information given or general knowledge, narrowing down the choices to the most reasonable one.
> > > > * Cluster 2 **Deductive Reasoning**: Focuses on using general principles or known facts to draw specific conclusions.
> > > > * Cluster 3 **Identifying Core Concepts**: Focuses on identifying the fundamental concepts or knowledge required to solve the problem.
> > > > * Cluster 4 **Analytical Reasoning**: Focuses on analyzing options or scenarios using critical thinking.

---

> > > > > ### Author Response · Authors · 2024-11-21
> > > > > **[5/5] Investigating Why ReGenesis Outperforms Competing Baselines**
> > > > >
> > > > > We show the clustering results of task-agnostic reasoning strategies on **OpenbookQA OOD inference** data generated by **ReGenesis**. (We only include examples for less straightforward cluster names due to the limit of the space.)
> > > > >
> > > > > **[ReGenesis] Clusters with examples:**
> > > > >
> > > > > * Cluster 1 **Process of Elimination**: This cluster involves strategies that use the process of elimination to systematically remove incorrect options until the correct one is identified.
> > > > > * Cluster 2 **Step-by-Step Reasoning**: This cluster includes strategies that focus on breaking down problems into smaller, manageable steps or components, allowing for a systematic evaluation and logical progression towards the solution.
> > > > > * Cluster 3 **Logical or Deductive Reasoning**: This cluster contains strategies that involve using logical or deductive reasoning to analyze problems, draw specific conclusions from general premises, and make well-informed decisions based on structured logical thinking.
> > > > > * Cluster 4 **Critical Thinking**: Strategies in this cluster involve critically analyzing each option, evaluating their implications and effectiveness, and considering external and internal factors to arrive at a logical conclusion based on evidence and reasoning.
> > > > > * Cluster 5 **Compare and Contrast or Evaluative Reasoning**: This cluster includes strategies that involve comparing and contrasting options based on specific criteria to evaluate their pros and cons and determine the best choice.
> > > > >    * **Question**: What has more gravity force than Earth but less than the sun? (A) Jupiter (B) the moon (C) a space station (D) a comet
> > > > >    * **ReGenesis Answer**: Identify the Objective: We need to find an object that has more gravity than Earth but less than the Sun. Define Known Values: - Earth's gravity: \(9.8 \text{ m/s}^2\) - Sun's gravity: \(274 \text{ m/s}^2\) Establish the Range: - We are looking for an object with a gravity force between Earth's and the Sun's gravity. Consider Possible Objects: - Jupiter, the moon, a space station, and a comet are all potential candidates. Evaluate Each Object: - Jupiter: Jupiter's gravity is approximately \(24.8 \text{ m/s}^2\). ... The Moon: The moon's gravity is approximately \(1.6 \text{ m/s}^2\). ... A Space Station: The gravity of a space station is negligible compared to Earth's, typically less than \(1 \text{ m/s}^2\). Therefore, it is .... A Comet: Comets generally have a gravity close to zero, so they do not have significant gravity compared to Earth or the Sun. Conclude the Evaluation: - Jupiter is the only object that satisfies the condition of having more gravity than Earth but less than the Sun. Verify the Outcome: - Double-checking the values, Jupiter's gravity is indeed \(24.8 \text{ m/s}^2\), ... Therefore, the object with more gravity than Earth but less than the Sun is (A) Jupiter.
> > > > >    * **LLM Summarized Strategy**: How can I systematically evaluate and compare each option to find the correct answer?
> > > > > * Cluster 6 **Use of Domain Knowledge or Common Knowledge**: Strategies in this cluster focus on leveraging existing knowledge, understanding of a subject, or common principles to answer questions accurately and confidently.
> > > > > * Cluster 7 **Systematic thinking and evaluation**: This approach involves identifying the problem and the parameters and systematically evaluating each option to eliminate incorrect answers and arrive at the correct choice.
> > > > >
> > > > >    * **Question**: What has more gravity force than Earth but less than the sun?(A) Jupiter (B) the moon (C) a space station (D) a comet
> > > > >     * **ReGenesis Answer**:  Jupiter: Jupiter is the largest planet in the solar system, with a mass much greater than Earth (about 318 times the mass of Earth). Its gravity on the surface is 24.79 m/s², which is more than Earth's gravity (9.81 m/s²), so it has more gravitational force than Earth. The Moon: The Moon's mass is only about 1/81 of Earth's, and its gravity is about 1/6th of Earth's gravity (1.62 m/s²). Thus, the Moon has far less gravity than Earth. A Space Station: A space station, like the International Space Station (ISS), is in orbit around Earth. The force of gravity at the ISS's altitude (about 400 km above Earth's surface) is still about 90% of the Earth's surface gravity. However, astronauts feel weightless due to the free-fall condition, not because there is no gravity. The gravitational force here is comparable to Earth's gravity, not more. A Comet: Comets are small, icy bodies in space with masses much smaller than Earth. The gravitational force they exert is much weaker than Earth's gravity. Therefore, a comet's gravity is much less than Earth's. Conclusion: Jupiter has more gravity than Earth and the Sun has much more. Therefore, Jupiter is the only option that has more gravity than Earth but less than the Sun. Correct Answer: (A) Jupiter
> > > > >     * **LLM Summarized Strategy**: How can I systematically evaluate and compare each option to find the correct answer?

---

> ### Author Response · Authors · 2024-11-21
> **Add one more Baseline: few-shot CoT prompting (w/o FT)**
>
> Reviewer: _”Missing some baselines: How will ReGenesis perform when compared with few-shot CoT prompting (w/o FT) on in-domain and out-of-domain tasks”_
>
> **Our response**:
>
> To address the concern, we introduce an additional baseline in Tables 2 and 3. This baseline utilizes 3-shot chain-of-thought (CoT) examples generated by GPT-4o, serving as a few-shot CoT prompting method without fine-tuning (w/o FT) using self-consistency. The results demonstrate that incorporating GPT-4o-generated CoT examples significantly improves the performance of Mistral-7B-Instruct-v0.3; however, it still lags behind ReGenesis on both in-domain and out-of-domain (OOD) tasks.
>
> | Training Methods | GSM8K | NumGLUE | ARC-c | ReClor | StrategyQA | ASDIV | SVAMP | AQUA  | BBH   | ANLI  | OpenBookQA |
> |------------------|-------|---------|-------|--------|------------|-------|-------|-------|-------|-------|------------|
> | w/o FT           | 60.0% | 38.2%   | 80.6% | 56.2%  | 80.8%      | 77.2% | 75.4% | 41.3% | 60.8% | 38.4% | 75.6%      |
> | **w/o FT (3-shot)**    | 68.3% | 47.2%   | 79.1% | 59.4%  | 80.8%      | 78.6% | 77.2% | 44.1% | 62.4% | 46.3% | 79.8%      |
> | StaR             | 66.0% | 64.5%   | 84.1% | 63.8%  | 85.9%      | 79.6% | 71.5% | 46.9% | 47.4% | 45.0% | 72.8%      |
> | ReGenesis        | 76.0% | 74.7%   | 85.4% | 70.6%  | 91.3%      | 81.2% | 83.9% | 48.8% | 69.3% | 49.5% | 81.4%      |

---

> > ### Author Response · Authors · 2024-11-21
> > **Mistral-7B Fine-Tuned on Gold Trajectories Falls Short of ReGenesis-Trained Model in Both In-Domain and OOD Scenarios**
> >
> > Reviewer: _”if Mistral-7B is fine-tuned on gold trajectories, which means removing the self-improvement setting, how large will the performance gap be between this and ReGenesis?”_
> >
> > **Our Response**:
> >
> > We use GSM8K’s provided golden trajectories to fine-tune Mistral-7B-Instruct-v0.3 and test the fine-tuned model on both in domain and OOD settings. Results are shown in the table below. From the results we’ve found that the model trained on GSM8K’s golden trajectories consistently underperforms the model trained with ReGenesis in in-domain and OOD settings given the same experiment settings, further showing the effectiveness of Regenesis.
> >
> > Table
> > | Training Methods               | GSM8K Test | ASDIV | SVAMP | AQUA  | BBH   | ANLI  | OpenBookQA |
> > |--------------------------------|------------|-------|-------|-------|-------|-------|------------|
> > | w/o FT                         | 60.0%      | 77.2% | 75.4% | 41.3% | 60.8% | 38.4% | 75.6%      |
> > | **FT with golden reasoning paths** | 68.3%      | 73.3% | 60.3% | 39.8% | 57.0% | 45.6% | 78.8%      |
> > | StaR                           | 66.0%      | 79.6% | 71.5% | 46.9% | 47.4% | 45.0% | 72.8%      |
> > | ReGenesis                      | 76.0%      | 81.2% | 83.9% | 48.8% | 69.3% | 49.5% | 81.4%      |

---

> > > ### Author Response · Authors · 2024-11-21
> > > **Explaining Why Providing Hints to LLMs During Final Reasoning Path Generation May Have Adverse Effects**
> > >
> > > Reviewer: _”Could the authors provide a detailed analysis and explanation as to why giving hints to models when generating reasoning paths might have adverse effects?”_
> > >
> > > **Our response**:
> > >
> > > The experiment results on the effect of hint locations are shown in Section 5.3 and Table 6. The results show that adding ground-truth answers as hints at the final reasoning path generation step is harmful in both in-domain and OOD settings. This is because the final reasoning path generation step also generates final answers given the reasoning paths since it is the final step of data generation. Therefore if in this step, we give the model the ground-truth answers as “hints”, we basically effectively change the final step into “fabricing the final reasoning path given the correct answer”. It is likely that the model would just copy-paste the provided correct answer to the end of its response, although its generated reasoning path is wrong and doesn’t lead to this answer. Therefore in this case, the later step of “filtering out good data by checking if the model’s final answer is the same as groundtruth answer” would be meaningless since even if the model’s final answer is correct, the reasoning path it has generated could be wrong and low quality.
> > >
> > > Here is an example in which if we provide the correct final answer in the final reasoning path generation step, the model just copy-paste the correct answer in the end while its synthesized reasoning path doesn’t lead to the correct answer.
> > >
> > > * **Example Question**: Mason opens the hood of his car and discovers that squirrels have been using his engine compartment to store nuts. If 2 busy squirrels have been stockpiling 30 nuts/day and one sleepy squirrel has been stockpiling 20 nuts/day, all for 40 days, how many nuts are in Mason's car?
> > >
> > > * **When we provide hint to the final reasoning path generation step, what we get**:  To determine the total number of nuts in Mason's car, let's follow the steps:\n\n1. Data Collection:\n   - Mason finds 2000 nuts in the car now (N0).\n\n2. Critical Thinking:\n   - We can identify two types of squirrels based on their activity levels: busy and sleepy.\n\n3. Systems Thinking and Analytical Thinking:\n   - The busy squirrels collect 30 nuts/day and the sleepy squirrel collects 20 nuts/day.\n\n4. Analytical Thinking:\n   - Calculate the total number of nuts collected by the busy squirrels over 40 days: Busy Squirrels Total = 30 * 40 = 1200 nuts\n   - Calculate the total number of nuts collected by the sleepy squirrel over 40 days: Sleepy Squirrel Total = 20 * 40 = 800 nuts\n   - Since the total number of nuts found in the car (N0) is less than the calculated total (2000 < 1200 + 800), there must be more nuts in the car.\n\n5. Evaluation:\n   - We need to add the nuts found in the car (N0) to the calculated total to find the final answer.\n   - Total nuts in the car = N0 + (Busy Squirrels Total + Sleepy Squirrel Total) = 2000 + (1200 + 800) = 3200 nuts.\n\nYour Answer: \\boxed{3200} nuts are in Mason's car.
> > >
> > > * **Correct Answer**: 3200
> > >
> > > * **Analysis**: To derive the correct answer:3200, which is provided as a hint in the final reasoning path generation step, the LLM generates illogical reasoning paths. In this example, it  invents number 2000, to arrive at a correct answer: 3200.

---

> > > > ### Author Response · Authors · 2024-11-21
> > > > **The Impact of Simpler Datasets for Training and More Challenging Tasks for OOD Testing**
> > > >
> > > > Reviewer: _”StrategyQA and OpenbookQA are not challenging for LLMs nowadays. What would happen if training is done on StrategyQA and testing is performed on CommonsenseQA, a more challenging out-of-domain dataset? Furthermore, ASDIV, SVAMP, and AQUA are quite simple for GSM8K. If training is conducted on these datasets and testing is done on GSM8K, what would happen?“_ (Simpler data sets for training and harder tasks for OOD testing.)
> > > >
> > > > **Our Response**:
> > > >
> > > > If we use LLM’s original performance on different datasets as a relatively objective measurement of how difficult different datasets are, then among all the training datasets we considered, StrategyQA and ARC-c are the two easiest since LLM has the highest accuracy on them (77.4% and 77.2% separately  in first row Table 2). When we use them as training datasets and test on OOD tasks, we found that they lead to model performance improvement on much harder OOD tasks such as AQUA for math and ANLI for NLI. AQUA and ANLI are harder data sets compared to StrategyQA and ARC-c since LLM original performance on them without any further fine-tuning is much lower (41.3% and 38.4% separately). Please see **Table 3** for detailed numbers.

---

> > > > > ### Author Response · Authors · 2024-11-21
> > > > > **Explanation for our Typo**
> > > > >
> > > > > Reviewer: _”Why is A+S not included in Table 4?”_
> > > > >
> > > > > **Our Response**:
> > > > >
> > > > > We apologize for our typo. In Section 5.1, when we mention “A+S” in line 421, we meant to say “A+P”: which utilizes only “Guidance Adaptation” and “Reasoning Path Generation”. Because “P” is the final step that generates a complete final reasoning path leading to an answer and therefore it cannot be left out. “A+S” cannot generate final reasoning paths leading to final answers, which means the data generated by “A+S” are not qualified training data and cannot be fairly compared with the results of the other settings. Section 5.1 is meant to confirm the effectiveness of the middle steps (“A” and “S”) in the reasoning generation process of ReGenesis.

---

> ### Author Response · Authors · 2024-11-24
> **Gentle reminder that the discussion period closes soon**
>
> This is a gentle reminder that the discussion period closes soon. We have thoroughly addressed the concerns raised, provided detailed rationales, and conducted additional experiments and case studies to strengthen our work. We would be greatly encouraged if the reviewers either raised their scores or engaged in further discussion with us. Thanks a lot.

---

> > ### Comment · Reviewer_TzKs · 2024-11-25
> >
> > Thanks for your detailed response. I have adjusted my scores accordingly.

---

> > > ### Author Response · Authors · 2024-11-27
> > >
> > > Thank you for your feedback and for updating your score. We’re pleased the additional analysis addressed your concerns.

---

### Official Review · Reviewer_tjkA · 2024-10-27

**Soundness:** 3
**Presentation:** 3
**Contribution:** 3
**Rating:** 6
**Confidence:** 5

**Summary:**

The paper introduces a novel framework called ReGenesis. It aims to improve the reasoning capabilities of large language models (LLMs) without relying on external supervision. Unlike prior self-synthesizing techniques that generate task-specific reasoning paths, ReGenesis creates general task-agnostic reasoning paths, promoting broader applicability. ReGenesis begins with abstract general guidelines, adapts them to specific tasks, and generates reasoning paths, refining the model's capability on both in-domain and out-of-domain (OOD) tasks. Experiments show that ReGenesis significantly enhances LLM performance across various domains, demonstrating its effectiveness in developing reasoning generalists capable of transferring knowledge to new tasks.

**Strengths:**

ReGenesis introduces a structured approach for LLM self-improvement by creating task-agnostic, generalizable reasoning paths, which is a substantial departure from the task-specific paths in prior methods.

ReGenesis shows superior performance in OOD tasks, addressing a major limitation of existing self-synthesizing methods. The model's flexibility across multiple reasoning domains suggests broader applicability.

The authors conducted thorough evaluations across various datasets, including mathematics, logical reasoning, and common sense. This diversity reinforces the model’s effectiveness and robustness.

Detailed ablation studies on various components, filtering methods, and language models strengthen the framework's design choices, validating each element's contribution to ReGenesis’s success.

**Weaknesses:**

1. I believe the paper presents a valuable approach by establishing task-agnostic general reasoning guidelines. However, I am curious whether employing a more advanced model, such as GPT-4, to formulate these guidelines might yield better results than having them generated by the 7B model itself.

2. I think many benchmarks in Table 3 might not fully qualify as out-of-domain (OOD) tasks. Previous work has often used GSM8K’s training set as demonstrations for datasets like ASDIV and SVAMP, which lack dedicated training data, in in-context learning (ICL) setups. In my understanding, true OOD tasks should go beyond simply math, commonsense, or logical reasoning to include other tasks like code generation, summarization, etc.

3. I suggest adding an “average” column to both Table 2 and Table 3 for a clearer overview of the results.

**Questions:**

1. Many tables look weird, such as Table 20 and Table 21, which have no borders.

2. Please use vector graphics as much as possible.

---

> ### Author Response · Authors · 2024-11-21
> **Summary of Response**
>
> We appreciate Reviewer tjkA's detailed and insightful feedback. In response, we will make the following response:
>
> * **Figures and Tables**: We will revise the figures and tables as suggested, including adding borders and incorporating an Average Score column in Tables 2 and 3.
>
> * **Advanced Models for Reasoning Paths**: we conducted experiments **using more advanced models (GPT-4o) to generate reasoning paths** for training data. Notably, employing GPT-4o for this generating reasoning paths with ReGenesis resulted in significant performance improvements in both in-domain and out-of-domain tasks.
>
> * **More Out-of-Domain Experiments**: As per the reviewer's suggestion, we have also run additional OOD experiments on **code generation benchmarks (MBPP and HumanEval)**. The results, reported as pass@10, show that after using ReGenesis, performance on code generation tasks is improved over both LLMs (Mistral-7B-Instruct-v0.3) without fine-tuning and those trained with STaR.
>
> We believe these changes will address the reviewer's concerns and enhance the clarity of our work. Thanks.

---

> > ### Author Response · Authors · 2024-11-21
> > **Using GPT-4o to generate reasoning paths led to significant performance improvements in both in-domain and out-of-domain tasks.**
> >
> > Reviewer: _"However, I am curious whether employing a more advanced model, such as GPT-4, to formulate these guidelines might yield better results than having them generated by the 7B model itself."_
> >
> > **Our Response**:
> >
> > We follow Reviewer tjkA’s suggestion to test ReGenesis by employing more advanced models to generate reasoning paths as training data.  Given GSM8K, we use GPT-4o to generate reasoning paths with Regenesis as training data, following the exact same generation and filtering procedure as Regenesis in Table $2$ and $3$. Then we fine-tune _Mistral-7B-Instruct-v0.3_ model with this data and present results as follows:
> >
> > | Training Methods         | GSM8K test | ASDIV | SVAMP | AQUA  | BBH   | ANLI  | OpenBookQA |
> > |--------------------------|------------|-------|-------|-------|-------|-------|------------|
> > | w/o FT                   | 60.0%      | 77.2% | 75.4% | 41.3% | 60.8% | 38.4% | 75.6%      |
> > | StaR                     | 66.0%      | 79.6% | 71.5% | 46.9% | 47.4% | 45.0% | 72.8%      |
> > | ReGenesis                | 76.0%      | 81.2% | 83.9% | 48.8% | 69.3% | 49.5% | 81.4%      |
> > | ReGenesis w/ GPT-4 paths | 84.1%      | 84.1% | 86.2% | 53.5% | 69.4% | 54.5% | 82.2%      |
> >
> >
> > Using GPT-4o to generate reasoning paths led to significant performance improvements in both in-domain and out-of-domain tasks.

---

> > > ### Comment · Reviewer_tjkA · 2024-11-25
> > >
> > > Thank you for your reply. What about my concern for OOD tasks?

---

> ### Author Response · Authors · 2024-12-01
> **OOD Evaluation on Code Generation**
>
> We thank Reviewer tjkA for pushing for OOD results on more distant tasks such as code generation. We apologize for the delayed response due to our limited GPU resources in the ICLR rebuttal season.
>
> We agree that including broader OOD tasks like code generation would enhance the evaluation. To address this, we conducted additional evaluations on two code generation benchmarks: MBPP and HumanEval. The results, reported as pass@10, show that **after using ReGenesis, performance on code generation tasks is improved over both LLMs (_Mistral-7B-Instruct-v0.3_) without fine-tuning and those trained with STaR**. We will include the results in our revised version.
>
> | Training Datasets |  Training Methods | MBPP       | HumanEval  |
> |-------------------|-------------------|------------|------------|
> | -                 | w/o FT            |      58.8% |      62.2% |
> | GSM8K             | STaR              |      58.7% |      49.5% |
> | GSM8K             | ReGenesis         |  **59.3%** |  **67.2%** |
> |                   |                   |            |            |
> | -                 | w/o FT            |      58.8% |      62.2% |
> | NumGLUE           | STaR              |      57.5% |      59.9% |
> | NumGLUE           | ReGenesis         | **59.30%** |  **63.5%** |
> |                   |                   |            |            |
> | -                 | w/o FT            |     58.80% |     62.20% |
> | ARC               | STaR              |     58.60% |     62.10% |
> | ARC               | ReGenesis         | **59.20%** | **64.10%** |
>
> We hope that these results effectively address the reviewer's concerns. Thanks so much.

---

### Official Review · Reviewer_KRUg · 2024-10-31

**Soundness:** 3
**Presentation:** 3
**Contribution:** 3
**Rating:** 8
**Confidence:** 4

**Summary:**

The paper introduces a method named ReGenesis to enhance the reasoning abilities of LLMs by allowing them to self-synthesize reasoning paths without human intervention or task-specific examples. Unlike prior methods that struggle with OOD generalization due to reliance on specific task examples, ReGenesis employs a progressive approach that builds from abstract reasoning guidelines to task-specific structures and paths. This approach encourages generalization across tasks, improving performance both in-domain and on OOD reasoning tasks.

Key contributions include:

1. A novel framework that allows LLMs to self-improve by generating diverse, task-agnostic reasoning paths.
2. Demonstrated effectiveness of ReGenesis on multiple datasets, with notable OOD performance improvements over existing methods.
3. An extensive analysis showing ReGenesis's adaptability across various LLMs, design configurations, and reasoning tasks.

**Strengths:**

1. The work provides a fresh insight into a key limitation in reasoning synthesis for LLMs, observing that existing self-synthesized reasoning methods often experience substantial performance drops in out-of-domain (OOD) settings.
2. The authors propose a novel abstract-to-concrete synthesis route, progressively transitioning from general reasoning guidelines to task-specific reasoning paths.
3. This paper presents comprehensive experiment, including extensive comparisons with competitive baselines like LMSI and STaR.

**Weaknesses:**

1. Although the paper includes a comprehensive ablation study (Tables 4 and 5) to examine the components of the ReGenesis framework, the fundamental reasons behind the performance improvements remain ambiguous. The framework is designed to construct reasoning solutions that incorporate *general task-agnostic reasoning guidance*. However, it is unclear how the ReGenesis-generated reasoning chains retain this generalizability across tasks. According to Table 25, the generated reasoning chains appear more specific and detailed, including elements like *Reasoning Structures*. While this added detail likely contributes to performance gains, the paper lacks clear evidence on how such specific structures support generalization across diverse tasks rather than being overfitted to in-domain scenarios.


2.  While the paper presents ReGenesis as highly effective, there is limited discussion on cases where the framework may fail or yield lower-quality reasoning paths. Especially in cases of OOD tasks, which may vary significantly in structure or complexity, it is important to understand where and when ReGenesis might underperform or produce suboptimal reasoning paths.  For instance, in Table 3, when the model was fine-tuned on NumGLUE, it exhibited a performance drop on ASDIV, an OOD math reasoning task. This raises questions about the potential limitations of ReGenesis when the source and target tasks have significant differences in structure or reasoning requirements.

**Questions:**

1. Could the authors clarify whether the process of editing and expanding seed guidelines remains consistent across different models used in the experiments? Additionally, how sensitive is ReGenesis to the selection of inappropriate guidelines? For instance, if an unsuitable guideline is chosen, could this negatively impact the reliability of the final solution, particularly in OOD scenarios?
2. In Table 2, ReGenesis demonstrates a substantial performance improvement on GSM8K, surpassing other methods by over 1%—a margin comparable to the performance increase from LLaMA3.1 8B to LLaMA3.1 70B. Such a substantial gain warrants further investigation. Could the authors provide a deeper analysis of the underlying factors contributing to these performance gains?
3. The experiments in Table 3 are conducted by fine-tuning on one of five in-domain datasets at a time. Would mixed training across multiple in-domain datasets improve the generalization performance?

---

> ### Author Response · Authors · 2024-11-21
> **Summary of Response**
>
> We thank Reviewer KRUg for their insightful questions and comments, which have greatly contributed to improving our submission. Below, we summarize our responses to address your concerns:
>
> 1. **Task-Agnostic Reasoning Strategies - Why ReGenesis works**: we run additional experiments demonstrating potential reasons of ReGenesis's superior OOD performance. We found that although ReGenesis trains LLMs with the final task-specific reasoning paths, the diverse task-agnostic reasoning strategies employed by ReGenesis in earlier generation steps are actually effectively preserved in the final task-specific reasoning paths, enabling LLMs to detect and learn these strategies easily. Even when trained exclusively on the final reasoning paths, the LLM successfully learns such task-agnostic reasoning strategies and adopts them to OOD tasks.
>
> 2. **Sensitivity to Guideline Selection and robustness to low-quality reasoning paths**: We address the sensitivity of ReGenesis to the selection of inappropriate guidelines, as detailed in our response. Additionally, we provide a table illustrating how different training datasets exhibit slight variations in preferences for specific seed reasoning guidelines.
>
> 3. **Performance Comparison on GSM8K**: A comprehensive case study on why ReGenesis outperforms STaR on the GSM8k dataset suggests that ReGensis leverages its diverse answering strategies to meet GSM8k’s requirements effectively.
>
> 4. **Mixing Training Datasets**: As shown in Appendix A.1, combining different training datasets slightly improves the LLM’s out-of-distribution performance.
>
> We appreciate the opportunity to further clarify and refine our work based on your valuable feedback.

---

> ### Author Response · Authors · 2024-11-21
> **[1/5] Task-Agnostic Reasoning Strategies Utilized by ReGenesis are Effectively Retained in Task-Specific Reasoning Paths, Maintaining Generalizability Across OOD Tasks**
>
> Reviewer: _“ReGenesis is designed to construct reasoning solutions that incorporate general task-agnostic reasoning guidance. However, it is unclear how the ReGenesis-generated reasoning chains retain this generalizability across tasks. According to Table 25, the generated reasoning chains appear more specific and detailed.” -> More analysis on why ReGenesis works for OOD tasks”_
>
> **Our Response**:
>
> **Although ReGenesis uses the final detailed and task-specific reasoning paths as training data, the diverse task-agnostic reasoning strategies that Regenesis adopts to generate such final task-specific paths, are effectively preserved in the final task-specific paths such that it is easy for LLMs, such as mistral-7B-instruct, to detect them and learn them from the final task-specific paths.** We run the following analysis to verify this hypothesis.
>
>
> We randomly select 500 reasoning paths from ReGenesis and STaR’s generated data of _NumGLUE_ training data set. Note that the 500 reasoning paths of each method are all final task-specific reasoning paths. Then, we prompt _Mistral-7B-Instruct-v0.3_ to label each of the total 1000 reasoning paths the task-agnostic reasoning strategy behind the task-specific path. The goal of this experiment is to see whether from the task-specific path only, LLMs can already tell the task-agnostic strategy behind it and that ReGenesis’s task-specific paths have more diverse task-agnostic strategies. If so, it indicates that LLMs are likely to be able to capture the task-agnostic strategies behind the task-specific paths in the training data, even if they are trained only with  task-specific paths.
>
>
> Specifically, after _Mistral-7B-Instruct-v0.3_ labeling the task-agnostic reasoning strategies, we task Mistral-7B-Instruct-v0.3 to cluster the task-agnostic reasoning strategies generated by itself and suggest cluster names for both ReGenesis and STaR. We present the results as follows. **Results show that it is quite apparent for Mistral-7B-Instruct-v0.3 that the final task-specific paths generated by ReGenesis, have more diverse task-agnostic reasoning strategies behind them, compared to STaR  (More strategy clusters for ReGenesis and less for STaR).** Note that throughout this experiment, we don’t give Mistral-7B-Instruct-v0.3 any knowledge on how ReGenesis generates the final task-specific paths and what are the general reasoning strategies used by ReGenesis in the first two data generation steps. **On this condition, Mistral-7B-Instruct-v0.3 can still back-engineer and mine the diverse task-agnostic reasoning paths behind them, suggesting that diverse task-agnostic reasoning strategies used by ReGenesis are effectively preserved in the final task-specific reasoning paths, and that Mistral-7B-Instruct-v0.3 can detect them easily.**
>
> We have also run the same analysis at OOD inference on _OpenBookQA_, on models fine-tuned with ReGenesis and STaR separately on NumGLUE dataset to show that models trained with ReGenesis, actually learned to adopt more diverse task-agnostic reasoning strategies at inference in OOD settings. We present the results below. **Results suggest that models trained with ReGenesis’s task specific reasoning paths, have learned the diverse task-agnostic reasoning strategies that ReGenesis has adopted for path generation and can apply them on OOD tasks.**
>
>
> **_We show the clustering results on NumGLUE training data and OpenbookQA OOD inference data below separately._**

---

> > ### Author Response · Authors · 2024-11-21
> > **[2/5] Task-Agnostic Reasoning Strategies Utilized by ReGenesis are Effectively Retained in Task-Specific Reasoning Paths, Maintaining Generalizability Across OOD Tasks**
> >
> > We show the clustering results of task-agnostic reasoning strategies mined from **STaR's** **NumGLUE training data**. (We only include examples for less straightforward cluster names due to the limit of the space.)
> >
> > **[STaR] Clusters with examples:**
> >
> > * Cluster 1: **Direct Calculation and Simplification**
> > * Cluster 2: **Algebraic and Formula-based Approaches**
> >     * **Question**: Claire decides to go to a nearby animal farm that has a total of 19 animals. He is standing in a queue near the entry gate, from the gate he could only see legs of the animals in the farm and he counted the total number of legs to be 56. He already knows that there are only ducks and horses in this farm. From this information find the number of ducks
> >     * **STaR Answer**: Let's say the number of ducks is D and the number of horses is H.\n\nSince each duck has 2 legs and each horse has 4 legs, the total number of legs can be represented by the equation:\n\n2D + 4H = 56\n\nSince there are 19 animals in total, we can set up another equation:\n\nD + H = 19\n\nWe can solve this system of equations by substituting D with 19 - H, which gives us:\n\n2(19 - H) + 4H = 56\n\nExpanding the equation and simplifying, we get:\n\n38 - 2H + 4H = 56\n2H = 18\nH = 9\n\nNow that we know the number of horses (H = 9), we can find the number of ducks (D) by substituting H into the first equation:\n\n2D + 4(9) = 56\n2D + 36 = 56\n2D = 20\nD = 10\n\nSo, there are 10 ducks in the farm.
> >     * **LLM Summarized Strategy**: How can I create equations that represent the problem and use them to find a solution?
> >
> > * Cluster 3: **Stoichiometry and Chemical Reactions**
> >
> >     * **Question**: How many moles of CH4 are required to react with 2 moles of Cl2 to form 2 moles of CH3Cl along with 2 moles of HCl
> >     * **STaR Answer**: The equation for the reaction is:\n\nCH4 (g) + Cl2 (g) \u2192 CH3Cl (g) + HCl (g)\n\nFrom the equation, we can see that:\n\n1 mole of CH4 reacts with 1 mole of Cl2 to form 1 mole of CH3Cl and 1 mole of HCl\n\nSince the question asks for 2 moles of CH3Cl and 2 moles of HCl, we can see that 2 moles of CH4 are required to react with 2 moles of Cl2 to form 2 moles of CH3Cl and 2 moles of HCl.\n\nTherefore, the answer is 2 moles of CH4.
> >     * **LLM Summarized Strategy**: Use direct application of stoichiometry: Identify the chemical reaction, balance it if necessary, and use the stoichiometric coefficients to directly calculate the required amount of reactants or products based on the given data.
> >
> > * Cluster 4: **Problem Decomposition and Step-by-Step Calculation**
> > * Cluster 5: **Logical Reasoning and Pattern Recognition**

---

> > > ### Author Response · Authors · 2024-11-21
> > > **[3/5] Task-Agnostic Reasoning Strategies Utilized by ReGenesis are Effectively Retained in Task-Specific Reasoning Paths, Maintaining Generalizability Across OOD Tasks**
> > >
> > > We show the clustering results of task-agnostic reasoning strategies mined from **ReGenesis's** **NumGLUE training data** . (We only include examples for less straightforward cluster names due to the limit of the space.)
> > >
> > > **[ReGenesis] Clusters with examples**:
> > >
> > > * Cluster 1: **Step-by-Step Process**
> > > * Cluster 2: **Applying Logic and Formulas**
> > > * Cluster 3: **Reflective Thinking**
> > >     * **Question**: Joshua became headmaster of a school in the month of May. Joshua stayed headmaster for 2 months before retiring in the same year. In which month did Joshua retire?
> > >     * **ReGenesis Answer**:
> > > Step 1: Reflect on initial biases and assumptions
> > > I recognize that the common Western calendar year structure consists of 12 months, but I should be open to alternative structures and month lengths.
> > > Step 2: Reflect on past experiences and learning.
> > > I've encountered problems that required creative thinking and consideration of different scenarios and structures. I'm prepared to think creatively and consider alternative month lengths or school calendar systems.
> > > Step 3: Reframe the problem and ask specific and open-ended questions.
> > > We know that Joshua became headmaster in May ...
> > > Step 4: Analyze and evaluate the problem.
> > > Joshua was headmaster for 2 months, and we know he retired ...
> > > Step 5: Make a decision or solution.
> > > Based on the analysis, we can conclude that Joshua retired in July, considering the Western calendar structure, or July or August considering alternative month lengths and school calendar systems.
> > > Final Answer: \\boxed{July}
> > >
> > >     * **LLM Summarized Strategy**:  Reflective Thinking: This strategy emphasizes stepping back from the problem, engaging in introspection and self-reflection, examining personal biases and assumptions, and considering alternative interpretations or structures.
> > >
> > > * Cluster 4: **Sequential and Chronological Analysis**
> > > * Cluster 5: **Information Extraction**
> > > * Cluster 6: **Problem Decomposition**
> > > * Cluster 7: **Systematic Thinking**
> > >     * **Question**: How many moles of Sodium bicarbonate are required to react with 3 moles of Hydrochrolic acid to form 3 moles of Water, 3 moles of Carbon dioxide and 3 moles of Sodium chloride
> > >     * **ReGenesis Answer**: Step 1: Define the System. Reactants: 3 moles of Hydrochloric acid (HCl) and x moles of Sodium bicarbonate (NaHCO3). Products: 3 moles of Water (H2O), 3 moles of Carbon dioxide (CO2), and 3 moles of Sodium chloride (NaCl). Step 2: Identify the Reaction. Chemical reaction between HCl and NaHCO3 to form H2O, CO2, and NaCl. Reactants: 3 moles of HCl and x moles of NaHCO3. Products: 3 moles of H2O, 3 moles of CO2, and 3 moles of NaCl. Step 3: Analyze Feedback Loops and Interdependencies. The reaction is irreversible, meaning reactants are consumed and cannot be reversed. Formation of H2O and CO2 affects reaction rate and NaCl formation. Formation of NaCl affects reaction rate and H2O and CO2 formation. Step 4: Develop a Holistic Solution. Set up a balanced chemical equation:
> > > 3 HCl + NaHCO3 → 3 H2O + 3 CO2 + NaCl. Step 5: Determine the Required Amount of NaHCO3. 1 mole of NaHCO3 reacts with 1 mole of HCl to form 1 mole of each product. To react 3 moles of HCl, 3 moles of NaHCO3 are required. Conclusion: 3 moles of Sodium bicarbonate (NaHCO3) are required to react with 3 moles of Hydrochloric acid (HCl) to form 3 moles of Water, 3 moles of Carbon dioxide, and 3 moles of Sodium chloride.
> > > Therefore, the answer is: \\boxed{3 moles} of Sodium bicarbonate are required.
> > >
> > >     * **LLM Summarized Strategy**: Use systems thinking: Consider the problem as part of a larger system and understanding the interconnectedness of various elements. Focuses on identifying the underlying causes, feedback loops, and interdependencies that influence the problem, and developing holistic solutions that address the system as a whole.
> > >
> > > * Cluster 8: **Questioning Assumptions and Critical Thinking**
> > >     * **Question**: Piper counted wings of all commercial planes placed in an air exhibition. If total wings were 90, how many planes were there?
> > >     * **ReGenesis Answer**: I. Establish the known facts: Total number of wings = 90. Each plane has two wings. II. Question initial assumption: Is the problem asking us to find the total number of wings or the number of planes? III. Challenge and revise initial assumption: If each plane has two wings, then we can set up an equation: 2x = 90, where x is the number of planes. Divide both sides by 2 to solve for x: x = 90 ÷ 2 = 45. IV. Conclusion: Therefore, the correct answer is that there are \\boxed{45} planes in the air exhibition.
> > >   * **LLM Summarized Strategy**: Start by establishing known facts, questioning assumptions, and setting up a simple equation based on logical deduction. Simplify the problem by considering the relationship between the number of wings and planes, and solve for the unknown variable.
> > > * Cluster 9: **Mathematical and Analytical Calculation**
> > > * Cluster 10: **Stoichiometry and Chemical Problem Solving**

---

> ### Author Response · Authors · 2024-11-21
> **[4/5] Task-Agnostic Reasoning Strategies Utilized by ReGenesis are Effectively Retained in Task-Specific Reasoning Paths, Maintaining Generalizability Across OOD Tasks**
>
> We show the clustering results of task-agnostic reasoning strategies mined from **STaR's** **OpenbookQA OOD inference** data. (We only include examples for less straightforward cluster names due to the limit of the space.)
>
> **[STaR] Clusters:**
> * Cluster 1 **Process of Elimination**: Focuses on systematically ruling out incorrect options to identify the correct answer.
>     * **Question**: Where might a bunny live? (A) a thicket (B) atop palm trees (C) a sewer system (D) a deserted island
>     * **STaR Answer**: A bunny would most likely live in a thicket, which is a dense growth of bushes or undergrowth, providing shelter and food. The other options are unlikely, as palm trees are tall and not typically found in habitats suitable for bunnies, sewer systems are not a suitable habitat for any animal, and a deserted island would be too isolated and lack the necessary food and shelter for a bunny. Therefore, the answer is (A).
>     * **LLM Summarized Strategy**: Process of elimination: Assess each potential answer choice and eliminate those that are clearly incorrect or implausible. This strategy involves evaluating the likelihood or suitability of each option based on the information given or general knowledge, narrowing down the choices to the most reasonable one.
> * Cluster 2 **Deductive Reasoning**: Focuses on using general principles or known facts to draw specific conclusions.
> * Cluster 3 **Identifying Core Concepts**: Focuses on identifying the fundamental concepts or knowledge required to solve the problem.
> * Cluster 4 **Analytical Reasoning**: Focuses on analyzing options or scenarios using critical thinking.

---

> ### Author Response · Authors · 2024-11-21
> **[5/5] Task-Agnostic Reasoning Strategies Utilized by ReGenesis are Effectively Retained in Task-Specific Reasoning Paths, Maintaining Generalizability Across OOD Tasks**
>
> We show the clustering results of task-agnostic reasoning strategies mined from **ReGenesis's** **OpenbookQA OOD inference** data. (We only include examples for less straightforward cluster names due to the limit of the space.)
>
> **[ReGenesis] Clusters with examples:**
>
> * Cluster 1 **Process of Elimination**: This cluster involves strategies that use the process of elimination to systematically remove incorrect options until the correct one is identified.
> * Cluster 2 **Step-by-Step Reasoning**: This cluster includes strategies that focus on breaking down problems into smaller, manageable steps or components, allowing for a systematic evaluation and logical progression towards the solution.
> * Cluster 3 **Logical or Deductive Reasoning**: This cluster contains strategies that involve using logical or deductive reasoning to analyze problems, draw specific conclusions from general premises, and make well-informed decisions based on structured logical thinking.
> * Cluster 4 **Critical Thinking**: Strategies in this cluster involve critically analyzing each option, evaluating their implications and effectiveness, and considering external and internal factors to arrive at a logical conclusion based on evidence and reasoning.
> * Cluster 5 **Compare and Contrast or Evaluative Reasoning**: This cluster includes strategies that involve comparing and contrasting options based on specific criteria to evaluate their pros and cons and determine the best choice.
>    * **Question**: What has more gravity force than Earth but less than the sun? (A) Jupiter (B) the moon (C) a space station (D) a comet
>    * **ReGenesis Answer**: Identify the Objective: We need to find an object that has more gravity than Earth but less than the Sun. Define Known Values: - Earth's gravity: \(9.8 \text{ m/s}^2\) - Sun's gravity: \(274 \text{ m/s}^2\) Establish the Range: - We are looking for an object with a gravity force between Earth's and the Sun's gravity. Consider Possible Objects: - Jupiter, the moon, a space station, and a comet are all potential candidates. Evaluate Each Object: - Jupiter: Jupiter's gravity is approximately \(24.8 \text{ m/s}^2\). ... The Moon: The moon's gravity is approximately \(1.6 \text{ m/s}^2\). ... A Space Station: The gravity of a space station is negligible compared to Earth's, typically less than \(1 \text{ m/s}^2\). Therefore, it is .... A Comet: Comets generally have a gravity close to zero, so they do not have significant gravity compared to Earth or the Sun. Conclude the Evaluation: - Jupiter is the only object that satisfies the condition of having more gravity than Earth but less than the Sun. Verify the Outcome: - Double-checking the values, Jupiter's gravity is indeed \(24.8 \text{ m/s}^2\), ... Therefore, the object with more gravity than Earth but less than the Sun is (A) Jupiter.
>    * **LLM Summarized Strategy**: How can I systematically evaluate and compare each option to find the correct answer?
> * Cluster 6 **Use of Domain Knowledge or Common Knowledge**: Strategies in this cluster focus on leveraging existing knowledge, understanding of a subject, or common principles to answer questions accurately and confidently.
> * Cluster 7 **Systematic thinking and evaluation**: This approach involves identifying the problem and the parameters and systematically evaluating each option to eliminate incorrect answers and arrive at the correct choice.
>
>    * **Question**: What has more gravity force than Earth but less than the sun?(A) Jupiter (B) the moon (C) a space station (D) a comet
>     * **ReGenesis Answer**:  Jupiter: Jupiter is the largest planet in the solar system, with a mass much greater than Earth (about 318 times the mass of Earth). Its gravity on the surface is 24.79 m/s², which is more than Earth's gravity (9.81 m/s²), so it has more gravitational force than Earth. The Moon: The Moon's mass is only about 1/81 of Earth's, and its gravity is about 1/6th of Earth's gravity (1.62 m/s²). Thus, the Moon has far less gravity than Earth. A Space Station: A space station, like the International Space Station (ISS), is in orbit around Earth. The force of gravity at the ISS's altitude (about 400 km above Earth's surface) is still about 90% of the Earth's surface gravity. However, astronauts feel weightless due to the free-fall condition, not because there is no gravity. The gravitational force here is comparable to Earth's gravity, not more. A Comet: Comets are small, icy bodies in space with masses much smaller than Earth. The gravitational force they exert is much weaker than Earth's gravity. Therefore, a comet's gravity is much less than Earth's. Conclusion: Jupiter has more gravity than Earth and the Sun has much more. Therefore, Jupiter is the only option that has more gravity than Earth but less than the Sun. Correct Answer: (A) Jupiter
>     * **LLM Summarized Strategy**: How can I systematically evaluate and compare each option to find the correct answer?

---

> > ### Author Response · Authors · 2024-11-21
> > **Different training data sets have slightly different preferences on different seed reasoning guidelines**
> >
> > Reviewer: _“limited discussion on cases where the framework may fail or yield lower-quality reasoning paths. Especially in cases of OOD tasks, which may vary significantly in structure or complexity, it is important to understand where and when ReGenesis might underperform or produce suboptimal reasoning paths. ”Could the authors clarify whether the process of editing and expanding seed guidelines remains consistent across different models used in the experiments? Additionally, how sensitive is ReGenesis to the selection of inappropriate guidelines? For instance, if an unsuitable guideline is chosen, could this negatively impact the reliability of the final solution, particularly in OOD scenarios?_
> >
> > **Our Response**:
> >
> > The process of editing and expanding seed guidelines remain consistent across all experiments in this paper submission. While we note that after generating reasoning paths and answers with the given LLMs, we use ground-truth answers to filter and keep the reasoning paths that lead to correct answers as training data, same as our baseline methods. This filtering process potentially serves as an auto-judge to automatically select proper seed guidance for a training data set, since seed guidances that lead to more correct answers will contribute more training data to the entire training set after filtering. In our analysis we observed that **different training data sets actually have slightly different preferences on different seed guidelines**,  in the sense that after ground-truth filtering, the numbers of paths left for seed guidelines are slightly different for different training data sets.We present the five most representative guidelines along with their corresponding percentages in the filtered correct reasoning paths.
> >
> > | Guidelines                 | GSM8K     | NumGLUE   | ARC_c     | ReClor    | StrategyQA |
> > |----------------------------|-----------|-----------|-----------|-----------|-------------|
> > | How can I break down...    | **4.62%** | **4.11%** | **6.40%** | 3.07%     | 4.32%       |
> > | Try creative thinking ...  | 1.75%     | 3.71%     | 2.41%     | 3.81%     | 3.33%       |
> > | Critical Thinking...       | 4.11%     | 4.09%     | 3.78%     | **5.12%** | **4.72%**   |
> > | Use Reflective Thinking... | 3.69%     | 3.98%     | 1.69%     | 3.32%     | 3.97%       |
> > | Use systems thinking...    | 3.41%     | 3.94%     | 3.05%     | 3.94%     | 4.45%       |
> >
> > The analysis reveals that for datasets like GSM8K, NumGLUE, and StrategyQA, the distribution of correct reasoning paths generated by our guideline framework is more uniform, resulting in a more diverse dataset. This diversity enables our approach to outperform Star, as these datasets demand varied reasoning strategies. For ARC-c, since Star primarily employs a "step-by-step question breakdown" strategy—which aligns closely with one of our generated reasoning paths—our performance is more comparable to STaR's.

---

> > > ### Author Response · Authors · 2024-11-21
> > > **[1/2] ReGenesis Shows Significant Performance Improvement on GSM8K Due to Its Diverse Answering Strategies, Perfectly Aligning with GSM8K's Needs.**
> > >
> > > Reviewer: _”In Table 2, ReGenesis demonstrates a substantial performance improvement on GSM8K, surpassing other methods by over 1%—a margin comparable to the performance increase from LLaMA3.1 8B to LLaMA3.1 70B. Such a substantial gain warrants further investigation. Could the authors provide a deeper analysis of the underlying factors contributing to these performance gains?”_
> > >
> > > **Our Response**:
> > >
> > > We **first** conducted a **case study** focusing on questions where our method succeeded but STaR failed. Our analysis revealed that our method employs much  more diverse strategies, which are absent in LLMs trained with STaR.
> > >
> > > 1. ReGenesis answers questions by **systematically** breaking down the problem into smaller parts: gathering data, calculating potential increases, and comparing profits. This approach demonstrates critical thinking and analytical reasoning, effectively leading to the correct conclusion.
> > >   * **Example question**: A merchant wants to make a choice of purchase between 2 purchase plans: jewelry worth 5,000 or electronic gadgets worth 8,000. His financial advisor speculates that the jewelry market will go up 2.5% while the electronic gadgets market will rise 1.2% within the same month. If the merchant is looking to maximize profit at the end of this month by making a choice, how much profit would this be?
> > >   * **STaR answer**: Let's calculate the future value of both options: 1. If the merchant chooses jewelry worth 5,000, the future value will be: 5,000 + (0.025 * 5,000) = <<5000 + (0.025 * 5000)=525>>5,250.  2. If the merchant chooses electronic gadgets worth 8,000, the future value will be: 8,000 + (0.012 * 8,000) = <<8000 + (0.012 * 8000)=808>>8,080. Comparing both values, the merchant should choose electronic gadgets worth $8,000 since it has a higher future value of 8,080 compared to jewelry worth 5,250. The profit the merchant would make is 8,080 - 8,000 = <<8080-8000=80>>80. #### 80
> > >   * **ReGenesis Answer**: 1. Gather data: - Current prices: 5,000 (jewelry) and 8,000 (electronic gadgets) - Expected market growth rates: 2.5% for jewelry and 1.2% for electronic gadgets 2. Calculate the potential increase in value: - For jewelry: 2.5% of 5,000 = 125 - For electronic gadgets: 1.2% of 8,000 = 96 3. Calculate the potential profit for each option: - Profit for jewelry: 125 (increased value) - Profit for electronic gadgets: 96 (increased value) 4. Compare the potential profits: - The merchant will make a profit of 125 if they choose jewelry, or $96 if they choose electronic gadgets. 5. Evaluate other factors: - The merchant should consider additional factors such as personal preferences, budget, and other financial obligations when making the final decision. 6. Make a decision: - Based on the calculated profits, the merchant will make a higher profit by choosing jewelry. However, they should weigh this with their personal financial situation and other factors before making the final decision. The merchant will make a profit of 125 if they choose jewelry.
> > >
> > > 2. ReGenesis also uses more **logical reasoning processes**, which involves identifying a pattern.
> > >   * **Example question**: Alice likes to count the puffs of clouds in the sky while she eats her lunch outside at school. On Monday she counts just 3 puffs of clouds. Each day after that through Friday, though, she sees double the number of clouds in the sky as the day before. At the end of the week, how many clouds will she have counted in the sky at lunch across all five days?
> > >   * **STaR answer**: On Monday, Alice counts 3 puffs of clouds. Tuesday: 3 x 2 = <<3*2=6>>6 puffs. Wednesday: 6 x 2 = <<6*2=12>>12 puffs. Thursday: 12 x 2 = <<12*2=24>>24 puffs. Friday: 24 x 2 = <<24*2=48>>48 puffs. Adding all of them up: 3 + 6 + 12 + 24 + 48 = <<3+6+12+24+48=83>>83 puffs. #### 83
> > >   * **ReGenesis Answer**: 1. Identify the initial number of clouds Alice sees on Monday: 3 clouds. 2. Recall that each day after Monday, she sees double the number of clouds from the previous day. 3. Create a pattern for the number of clouds she sees on each day: Monday: 3 clouds; Tuesday: 3 * 2 = 6 clouds; Wednesday: 6 * 2 = 12 clouds; Thursday: 12 * 2 = 24 clouds; Friday: 24 * 2 = 48 clouds. 4. Add up the number of clouds she sees each day to find the total number of clouds across the week: Total clouds = Monday + Tuesday + Wednesday + Thursday + Friday; Total clouds = 3 + 6 + 12 + 24 + 48; Total clouds = 93 clouds. So, Alice will have counted 93 clouds in the sky at lunch across all five days.

---

> > > > ### Author Response · Authors · 2024-11-21
> > > > **[2/2] ReGenesis Shows Significant Performance Improvement on GSM8K Due to Its Diverse Answering Strategies, Perfectly Aligning with GSM8K's Needs.**
> > > >
> > > > 3. ReGenesis also conducts more **critical thinking** to make the answer correct.
> > > > * **Example Question**: Lee used to be able to run the 400-meter hurdles two seconds faster than Gerald would run the 400-meter hurdles.  But Gerald changed his diet, which improved his speed by 10%.  If Lee runs the 400-meter hurdles in 38 seconds, how fast can Gerald, with his improved diet, run the 400-meter hurdles, in seconds?
> > > > * **STaR answer**: Since Lee runs 2 seconds faster than Gerald, we can say that when they were running with their previous diets, Lee ran in t seconds and Gerald ran in t + 2 seconds, where t represents the time in seconds that Lee ran with his improved diet (which we don't know yet).
> > > > Their respective times in seconds with their old diets were related by: t + 2 = (t - 2) * 0.9 (Gerald's time increased by 10%). Solving for t: t = (2 / (1 - 0.9)) + 2 = 2 / 0.1 + 2 = <<2 / 0.1 + 2 = 20 + 2 = 22>>22.
> > > > Since Lee runs the 400-meter hurdles in 38 seconds with his improved diet, we know that: t = 38.
> > > > So, with his improved diet, Gerald can run the 400-meter hurdles in t + 2 = 38 + 2 = <<38+2=40>>40 seconds. #### 40
> > > > * **ReGenesis Answer**: 1. Establish the initial time difference between Lee and Gerald: - We know that Lee runs the 400-meter hurdles in 38 seconds, and Gerald used to be 2 seconds slower. - Therefore, Gerald’s initial time for the 400-meter hurdles was \(38 + 2 = 40\) seconds. 2. Account for the 10% improvement in Gerald’s speed: - A 10% improvement in speed means that Gerald's running time will decrease, as he will now run faster. - To calculate the new time, we need to find 90% of his original time (since he is now 10% faster, the remaining 90% represents his new time). - The formula for the new time is: \[ \text{New Time} = \text{Initial Time} \times 0.90 \] - Substituting the values: \[ \text{New Time} = 40 \times 0.90 = 36 \text{ seconds} \] Therefore, after improving his diet, Gerald can now run the 400-meter hurdles in 36 seconds, reflecting a 10% improvement in his speed.
> > > >
> > > > 4. ReGenesis also uses **reflective thinking** by ensuring the calculations align with the task requirement
> > > > * **Example Question**:Lee used to be able to run the 400-meter hurdles two seconds faster than Gerald would run the 400-meter hurdles.  But Gerald changed his diet, which improved his speed by 10%.  If Lee runs the 400-meter hurdles in 38 seconds, how fast can Gerald, with his improved diet, run the 400-meter hurdles, in seconds?
> > > > * **STaR answer**: Let's call the initial time that Gerald could run the 400m hurdles before changing his diet as t. Lee could run 2 seconds faster, so his time is t - 2 seconds. Since the difference between their times is 2 seconds, we can write the following equation: t - (t - 2) = 38. Solving this equation gives t = 38 + 2 = <<38+2=40>>40. Before changing his diet, Gerald ran the 400m hurdles in 40 seconds. Gerald improved his speed by 10%, so his new time is: 40 * (1 + 0.1) = 40 * 1.1 = <<40\*1.1=44>>44 seconds. #### 44
> > > > * **ReGenesis Answer**: To find out how fast Gerald can run the 400-meter hurdles after improving his diet by 10%, we can follow these steps: 1. Recall the initial difference in running times between Lee and Gerald: Lee runs 400-meter hurdles in 38 seconds, and Gerald is 2 seconds slower. So, Gerald's initial time was 38 + 2 = 40 seconds. 2. Calculate the improvement in Gerald's time due to his diet change: Since the improvement is a 10% increase in speed, we can find the new time by subtracting 10% of the initial time. The formula for this is: New Time = Initial Time - (Initial Time * Percentage Change). In this case, New Time = 40 - (40 * 0.10) = 40 - 4 = 36 seconds. So, with his improved diet, Gerald can run the 400-meter hurdles in approximately 36 seconds.
> > > >
> > > > **Finally**, we evaluated the accuracy of _Mistral-7B-Instruct-v0.3_ prompted with various reasoning guidelines on the GSM8K dataset. The results indicate that no single strategy is effective for all questions, necessitating the use of multiple strategies. As demonstrated, LLMs trained with ReGenesis show a broader range of answering strategies compared to those trained with STaR. We attribute this enhanced performance to the diverse answering strategies learned by the ReGenesis-trained LLM from its training data.
> > > > | Guidelines                 | GSM8K  |
> > > > |----------------------------|--------|
> > > > | How can I break down...    | 53.45% |
> > > > | Try creative thinking ...  | 20.30% |
> > > > | Critical Thinking...       | 47.56% |
> > > > | Use Reflective Thinking... | 42.69% |
> > > > | Use systems thinking...    | 39.44% |

---

> > > > > ### Author Response · Authors · 2024-11-21
> > > > > **Mixing Different Training Datasets slightly improve LLMs’ OOD performance**
> > > > >
> > > > > Reviewer: _”The experiments in Table 3 are conducted by fine-tuning on one of five in-domain datasets at a time. Would mixed training across multiple in-domain datasets improve the generalization performance?”_
> > > > >
> > > > > **Our Response**:
> > > > >
> > > > > We conducted this analysis and included it in **Appendix A.1**.  The results show that multiple training datasets can slightly improve LLMs’ OOD performance. Please see details in Appendix A.1 and Figure 3.

---

> ### Comment · Reviewer_KRUg · 2024-11-21
> **Response to Author8842**
>
> Thank you very much for your detailed responses and extensive experiments. Your thorough explanations have resolved my doubts and addressed my concerns. I have increased my score accordingly.

---

> > ### Author Response · Authors · 2024-11-21
> >
> > Thank you so much for your kind words and for taking the time to review our work thoroughly. We're glad that our responses and experiments were able to clarify your concerns. We deeply appreciate your updated score. Please feel free to reach out with any further questions or suggestions!

---

### Official Review · Reviewer_w4BA · 2024-10-31

**Soundness:** 3
**Presentation:** 3
**Contribution:** 3
**Rating:** 8
**Confidence:** 4

**Summary:**

ReGenesis enhances LLMs' reasoning by self-synthesizing generalizable reasoning paths without human supervision, outperforming prior methods on in-domain and out-of-domain tasks. It improves OOD performance by 6.1%, while existing methods show a decline, proving its effectiveness across models and configurations.

**Strengths:**

- The problem of building a reasoning generalist is important, compared to other works focusing on dataset-specific improvements in LLMs.

- Comprehensive experiments highlight ReGenesis’s superior performance in both in-domain and out-of-domain tasks.

**Weaknesses:**

- The major concern is that there is a mismatch between the motivation and the proposed method. The paper pointed out that the bottleneck of self-synthesizing methods is that they suffer from poor generalization to out-of-domain (OOD), and hypothesized that it is because "their self-synthesized reasoning paths are too task-specific, lacking general task-agnostic reasoning guidance". However, even though they proposed a multi-level prompting method starting from general guidelines, in the end, they still used task-specific reasoning paths to train the model. While the authors suggest their method can enhance the diversity of reasoning chains, which helps with OOD performance, there is no clear analysis.

- The core challenge of creating a reasoning generalist may lie more in synthesizing diverse reasoning problems rather than just generating diverse reasoning chains. ReGenesis still relies on access to a dataset of questions and even ground-truth answers, limiting its potential for fully autonomous reasoning and diminishing its broader impact.

**Questions:**

- Can you compare the size of synthetic data you use for all methods? You mentioned that 32 reasoning paths are sampled for STaR, but how many of them are used for training after filtering?

- Why is there such a large difference between STaR and the proposed method on GSM8k? (46.3 vs 63.6)

---

> ### Author Response · Authors · 2024-11-21
> **Summary of Response**
>
> We deeply appreciate Reviewer w4BA’s insightful comments, which has guided us to add more analysis to strengthen our motivation and to show lights upon the potential reason why ReGenesis works. We respond to Reviewer w4BA’s comments as follows with more analytical results.
>
> * **Preservation of task-agnostic Reasoning Strategies**: We conducted experiments to demonstrate that the task-agnostic reasoning strategies used by ReGenesis are effectively preserved in the final task-specific reasoning paths, allowing LLMs to easily detect and learn them. Despite being trained only with the final task-specific reasoning paths generated by ReGenesis, LLMs successfully learn these task-agnostic strategies and generalize them to out-of-distribution (OOD) tasks.
> * **Data Size Consistency Across Models**: We present a table showing that the size of the synthetic data, after filtering, remains consistent across different baselines and our models.
> * **Performance Without Ground-Truth Data**: As shown in Table 5, ReGenesis outperforms the baseline models even without relying on ground-truth data.
> * **Case Study on GSM8k**: We also conducted a comprehensive case study, which reveals why ReGenesis significantly outperforms STaR on the GSM8k dataset.

---

> ### Author Response · Authors · 2024-11-21
> **[1/5] Task-agnostic reasoning strategies employed by ReGenesis are effectively preserved in the final task-specific reasoning paths, enabling LLMs to detect and learn them easily**
>
> Reviewer: _“Potential mismatch between the motivation and the proposed method. The authors hypothesized that current methods’ underperforms on OOD tasks because "their self-synthesized reasoning paths are too task-specific, lacking general task-agnostic reasoning guidance". However, even though they proposed a multi-level prompting method starting from general guidelines, in the end, they still used task-specific reasoning paths to train the model. While the authors suggest their method can enhance the diversity of reasoning chains, which helps with OOD performance, there is no clear analysis.”_
>
> **Our Response**:
>
> **Although ReGenesis uses the final detailed and task-specific reasoning paths as training data, the diverse task-agnostic reasoning strategies that Regenesis adopts to generate such final task-specific paths, are effectively preserved in the final task-specific paths such that it is easy for LLMs, such as mistral-7B-instruct, to detect them and learn them from the final task-specific paths.** We run the following analysis to verify this hypothesis.
>
>
> We randomly select 500 reasoning paths from ReGenesis and STaR’s generated data of _NumGLUE_ training data set. Note that the 500 reasoning paths of each method are all final task-specific reasoning paths. Then, we prompt _Mistral-7B-Instruct-v0.3_ to label each of the total 1000 reasoning paths the task-agnostic reasoning strategy behind the task-specific path. The goal of this experiment is to see whether from the task-specific path only, LLMs can already tell the task-agnostic strategy behind it and that ReGenesis’s task-specific paths have more diverse task-agnostic strategies. If so, it indicates that LLMs are likely to be able to capture the task-agnostic strategies behind the task-specific paths in the training data, even if they are trained only with  task-specific paths.
>
>
> Specifically, after _Mistral-7B-Instruct-v0.3_ labeling the task-agnostic reasoning strategies, we task Mistral-7B-Instruct-v0.3 to cluster the task-agnostic reasoning strategies generated by itself and suggest cluster names for both ReGenesis and STaR. We present the results as follows. **Results show that it is quite apparent for Mistral-7B-Instruct-v0.3 that the final task-specific paths generated by ReGenesis, have more diverse task-agnostic reasoning strategies behind them, compared to STaR (More strategy clusters for ReGenesis and less for STaR).** Note that throughout this experiment, we don’t give Mistral-7B-Instruct-v0.3 any knowledge on how ReGenesis generates the final task-specific paths and what are the general reasoning strategies used by ReGenesis in the first two data generation steps. **On this condition, Mistral-7B-Instruct-v0.3 can still back-engineer and mine the diverse task-agnostic reasoning paths behind them, suggesting that diverse task-agnostic reasoning strategies used by ReGenesis are effectively preserved in the final task-specific reasoning paths, and that Mistral-7B-Instruct-v0.3 can detect them easily.**
>
> We have also run the same analysis at OOD inference on _OpenBookQA_, on models fine-tuned with ReGenesis and STaR separately on NumGLUE dataset to show that models trained with ReGenesis, actually learned to adopt more diverse task-agnostic reasoning strategies at inference in OOD settings. We present the results below. **Results suggest that models trained with ReGenesis’s task specific reasoning paths, have learned the diverse task-agnostic reasoning strategies that ReGenesis has adopted for path generation and can apply them on OOD tasks.**
>
>
> **_We show the clustering results on NumGLUE training data and OpenbookQA OOD inference data below separately._**

---

> ### Author Response · Authors · 2024-11-21
> **[2/5] Task-agnostic reasoning strategies employed by ReGenesis are effectively preserved in the final task-specific reasoning paths, enabling LLMs to detect and learn them easily**
>
> We show the clustering results of task-agnostic reasoning strategies mined from **STaR's** **NumGLUE training data**. (We only include examples for less straightforward cluster names due to the limit of the space.)
>
> **[STaR] Clusters with examples:**
>
> * Cluster 1: **Direct Calculation and Simplification**
> * Cluster 2: **Algebraic and Formula-based Approaches**
>     * **Question**: Claire decides to go to a nearby animal farm that has a total of 19 animals. He is standing in a queue near the entry gate, from the gate he could only see legs of the animals in the farm and he counted the total number of legs to be 56. He already knows that there are only ducks and horses in this farm. From this information find the number of ducks
>     * **STaR Answer**: Let's say the number of ducks is D and the number of horses is H.\n\nSince each duck has 2 legs and each horse has 4 legs, the total number of legs can be represented by the equation:\n\n2D + 4H = 56\n\nSince there are 19 animals in total, we can set up another equation:\n\nD + H = 19\n\nWe can solve this system of equations by substituting D with 19 - H, which gives us:\n\n2(19 - H) + 4H = 56\n\nExpanding the equation and simplifying, we get:\n\n38 - 2H + 4H = 56\n2H = 18\nH = 9\n\nNow that we know the number of horses (H = 9), we can find the number of ducks (D) by substituting H into the first equation:\n\n2D + 4(9) = 56\n2D + 36 = 56\n2D = 20\nD = 10\n\nSo, there are 10 ducks in the farm.
>     * **LLM Summarized Strategy**: How can I create equations that represent the problem and use them to find a solution?
>
> * Cluster 3: **Stoichiometry and Chemical Reactions**
>
>     * **Question**: How many moles of CH4 are required to react with 2 moles of Cl2 to form 2 moles of CH3Cl along with 2 moles of HCl
>     * **STaR Answer**: The equation for the reaction is:\n\nCH4 (g) + Cl2 (g) \u2192 CH3Cl (g) + HCl (g)\n\nFrom the equation, we can see that:\n\n1 mole of CH4 reacts with 1 mole of Cl2 to form 1 mole of CH3Cl and 1 mole of HCl\n\nSince the question asks for 2 moles of CH3Cl and 2 moles of HCl, we can see that 2 moles of CH4 are required to react with 2 moles of Cl2 to form 2 moles of CH3Cl and 2 moles of HCl.\n\nTherefore, the answer is 2 moles of CH4.
>     * **LLM Summarized Strategy**: Use direct application of stoichiometry: Identify the chemical reaction, balance it if necessary, and use the stoichiometric coefficients to directly calculate the required amount of reactants or products based on the given data.
>
> * Cluster 4: **Problem Decomposition and Step-by-Step Calculation**
> * Cluster 5: **Logical Reasoning and Pattern Recognition**

---

> ### Author Response · Authors · 2024-11-21
> **[3/5] Task-agnostic reasoning strategies employed by ReGenesis are effectively preserved in the final task-specific reasoning paths, enabling LLMs to detect and learn them easily**
>
> We show the clustering results of task-agnostic reasoning strategies mined from **ReGenesis's** **NumGLUE training data** . (We only include examples for less straightforward cluster names due to the limit of the space.)
>
> **[ReGenesis] Clusters with examples**:
>
> * Cluster 1: **Step-by-Step Process**
> * Cluster 2: **Applying Logic and Formulas**
> * Cluster 3: **Reflective Thinking**
>     * **Question**: Joshua became headmaster of a school in the month of May. Joshua stayed headmaster for 2 months before retiring in the same year. In which month did Joshua retire?
>     * **ReGenesis Answer**:
> Step 1: Reflect on initial biases and assumptions
> I recognize that the common Western calendar year structure consists of 12 months, but I should be open to alternative structures and month lengths.
> Step 2: Reflect on past experiences and learning.
> I've encountered problems that required creative thinking and consideration of different scenarios and structures. I'm prepared to think creatively and consider alternative month lengths or school calendar systems.
> Step 3: Reframe the problem and ask specific and open-ended questions.
> We know that Joshua became headmaster in May ...
> Step 4: Analyze and evaluate the problem.
> Joshua was headmaster for 2 months, and we know he retired ...
> Step 5: Make a decision or solution.
> Based on the analysis, we can conclude that Joshua retired in July, considering the Western calendar structure, or July or August considering alternative month lengths and school calendar systems.
> Final Answer: \\boxed{July}
>
>     * **LLM Summarized Strategy**:  Reflective Thinking: This strategy emphasizes stepping back from the problem, engaging in introspection and self-reflection, examining personal biases and assumptions, and considering alternative interpretations or structures.
>
> * Cluster 4: **Sequential and Chronological Analysis**
> * Cluster 5: **Information Extraction**
> * Cluster 6: **Problem Decomposition**
> * Cluster 7: **Systematic Thinking**
>     * **Question**: How many moles of Sodium bicarbonate are required to react with 3 moles of Hydrochrolic acid to form 3 moles of Water, 3 moles of Carbon dioxide and 3 moles of Sodium chloride
>     * **ReGenesis Answer**: Step 1: Define the System. Reactants: 3 moles of Hydrochloric acid (HCl) and x moles of Sodium bicarbonate (NaHCO3). Products: 3 moles of Water (H2O), 3 moles of Carbon dioxide (CO2), and 3 moles of Sodium chloride (NaCl). Step 2: Identify the Reaction. Chemical reaction between HCl and NaHCO3 to form H2O, CO2, and NaCl. Reactants: 3 moles of HCl and x moles of NaHCO3. Products: 3 moles of H2O, 3 moles of CO2, and 3 moles of NaCl. Step 3: Analyze Feedback Loops and Interdependencies. The reaction is irreversible, meaning reactants are consumed and cannot be reversed. Formation of H2O and CO2 affects reaction rate and NaCl formation. Formation of NaCl affects reaction rate and H2O and CO2 formation. Step 4: Develop a Holistic Solution. Set up a balanced chemical equation:
> 3 HCl + NaHCO3 → 3 H2O + 3 CO2 + NaCl. Step 5: Determine the Required Amount of NaHCO3. 1 mole of NaHCO3 reacts with 1 mole of HCl to form 1 mole of each product. To react 3 moles of HCl, 3 moles of NaHCO3 are required. Conclusion: 3 moles of Sodium bicarbonate (NaHCO3) are required to react with 3 moles of Hydrochloric acid (HCl) to form 3 moles of Water, 3 moles of Carbon dioxide, and 3 moles of Sodium chloride.
> Therefore, the answer is: \\boxed{3 moles} of Sodium bicarbonate are required.
>
>     * **LLM Summarized Strategy**: Use systems thinking: Consider the problem as part of a larger system and understanding the interconnectedness of various elements. Focuses on identifying the underlying causes, feedback loops, and interdependencies that influence the problem, and developing holistic solutions that address the system as a whole.
>
> * Cluster 8: **Questioning Assumptions and Critical Thinking**
>     * **Question**: Piper counted wings of all commercial planes placed in an air exhibition. If total wings were 90, how many planes were there?
>     * **ReGenesis Answer**: I. Establish the known facts: Total number of wings = 90. Each plane has two wings. II. Question initial assumption: Is the problem asking us to find the total number of wings or the number of planes? III. Challenge and revise initial assumption: If each plane has two wings, then we can set up an equation: 2x = 90, where x is the number of planes. Divide both sides by 2 to solve for x: x = 90 ÷ 2 = 45. IV. Conclusion: Therefore, the correct answer is that there are \\boxed{45} planes in the air exhibition.
>   * **LLM Summarized Strategy**: Start by establishing known facts, questioning assumptions, and setting up a simple equation based on logical deduction. Simplify the problem by considering the relationship between the number of wings and planes, and solve for the unknown variable.
> * Cluster 9: **Mathematical and Analytical Calculation**
> * Cluster 10: **Stoichiometry and Chemical Problem Solving**

---

> ### Author Response · Authors · 2024-11-21
> **[4/5] Task-agnostic reasoning strategies employed by ReGenesis are effectively preserved in the final task-specific reasoning paths, enabling LLMs to detect and learn them easily**
>
> We show the clustering results of task-agnostic reasoning strategies mined from **STaR's** **OpenbookQA OOD inference** data. (We only include examples for less straightforward cluster names due to the limit of the space.)
>
> **[STaR] Clusters:**
> * Cluster 1 **Process of Elimination**: Focuses on systematically ruling out incorrect options to identify the correct answer.
>     * **Question**: Where might a bunny live? (A) a thicket (B) atop palm trees (C) a sewer system (D) a deserted island
>     * **STaR Answer**: A bunny would most likely live in a thicket, which is a dense growth of bushes or undergrowth, providing shelter and food. The other options are unlikely, as palm trees are tall and not typically found in habitats suitable for bunnies, sewer systems are not a suitable habitat for any animal, and a deserted island would be too isolated and lack the necessary food and shelter for a bunny. Therefore, the answer is (A).
>     * **LLM Summarized Strategy**: Process of elimination: Assess each potential answer choice and eliminate those that are clearly incorrect or implausible. This strategy involves evaluating the likelihood or suitability of each option based on the information given or general knowledge, narrowing down the choices to the most reasonable one.
> * Cluster 2 **Deductive Reasoning**: Focuses on using general principles or known facts to draw specific conclusions.
> * Cluster 3 **Identifying Core Concepts**: Focuses on identifying the fundamental concepts or knowledge required to solve the problem.
> * Cluster 4 **Analytical Reasoning**: Focuses on analyzing options or scenarios using critical thinking.

---

> ### Author Response · Authors · 2024-11-21
> **[5/5] Task-agnostic reasoning strategies employed by ReGenesis are effectively preserved in the final task-specific reasoning paths, enabling LLMs to detect and learn them easily**
>
> We show the clustering results of task-agnostic reasoning strategies mined from **ReGenesis's** **OpenbookQA OOD inference** data. (We only include examples for less straightforward cluster names due to the limit of the space.)
>
> **[ReGenesis] Clusters with examples:**
>
> * Cluster 1 **Process of Elimination**: This cluster involves strategies that use the process of elimination to systematically remove incorrect options until the correct one is identified.
> * Cluster 2 **Step-by-Step Reasoning**: This cluster includes strategies that focus on breaking down problems into smaller, manageable steps or components, allowing for a systematic evaluation and logical progression towards the solution.
> * Cluster 3 **Logical or Deductive Reasoning**: This cluster contains strategies that involve using logical or deductive reasoning to analyze problems, draw specific conclusions from general premises, and make well-informed decisions based on structured logical thinking.
> * Cluster 4 **Critical Thinking**: Strategies in this cluster involve critically analyzing each option, evaluating their implications and effectiveness, and considering external and internal factors to arrive at a logical conclusion based on evidence and reasoning.
> * Cluster 5 **Compare and Contrast or Evaluative Reasoning**: This cluster includes strategies that involve comparing and contrasting options based on specific criteria to evaluate their pros and cons and determine the best choice.
>    * **Question**: What has more gravity force than Earth but less than the sun? (A) Jupiter (B) the moon (C) a space station (D) a comet
>    * **ReGenesis Answer**: Identify the Objective: We need to find an object that has more gravity than Earth but less than the Sun. Define Known Values: - Earth's gravity: \(9.8 \text{ m/s}^2\) - Sun's gravity: \(274 \text{ m/s}^2\) Establish the Range: - We are looking for an object with a gravity force between Earth's and the Sun's gravity. Consider Possible Objects: - Jupiter, the moon, a space station, and a comet are all potential candidates. Evaluate Each Object: - Jupiter: Jupiter's gravity is approximately \(24.8 \text{ m/s}^2\). ... The Moon: The moon's gravity is approximately \(1.6 \text{ m/s}^2\). ... A Space Station: The gravity of a space station is negligible compared to Earth's, typically less than \(1 \text{ m/s}^2\). Therefore, it is .... A Comet: Comets generally have a gravity close to zero, so they do not have significant gravity compared to Earth or the Sun. Conclude the Evaluation: - Jupiter is the only object that satisfies the condition of having more gravity than Earth but less than the Sun. Verify the Outcome: - Double-checking the values, Jupiter's gravity is indeed \(24.8 \text{ m/s}^2\), ... Therefore, the object with more gravity than Earth but less than the Sun is (A) Jupiter.
>    * **LLM Summarized Strategy**: How can I systematically evaluate and compare each option to find the correct answer?
> * Cluster 6 **Use of Domain Knowledge or Common Knowledge**: Strategies in this cluster focus on leveraging existing knowledge, understanding of a subject, or common principles to answer questions accurately and confidently.
> * Cluster 7 **Systematic thinking and evaluation**: This approach involves identifying the problem and the parameters and systematically evaluating each option to eliminate incorrect answers and arrive at the correct choice.
>
>    * **Question**: What has more gravity force than Earth but less than the sun?(A) Jupiter (B) the moon (C) a space station (D) a comet
>     * **ReGenesis Answer**:  Jupiter: Jupiter is the largest planet in the solar system, with a mass much greater than Earth (about 318 times the mass of Earth). Its gravity on the surface is 24.79 m/s², which is more than Earth's gravity (9.81 m/s²), so it has more gravitational force than Earth. The Moon: The Moon's mass is only about 1/81 of Earth's, and its gravity is about 1/6th of Earth's gravity (1.62 m/s²). Thus, the Moon has far less gravity than Earth. A Space Station: A space station, like the International Space Station (ISS), is in orbit around Earth. The force of gravity at the ISS's altitude (about 400 km above Earth's surface) is still about 90% of the Earth's surface gravity. However, astronauts feel weightless due to the free-fall condition, not because there is no gravity. The gravitational force here is comparable to Earth's gravity, not more. A Comet: Comets are small, icy bodies in space with masses much smaller than Earth. The gravitational force they exert is much weaker than Earth's gravity. Therefore, a comet's gravity is much less than Earth's. Conclusion: Jupiter has more gravity than Earth and the Sun has much more. Therefore, Jupiter is the only option that has more gravity than Earth but less than the Sun. Correct Answer: (A) Jupiter
>     * **LLM Summarized Strategy**: How can I systematically evaluate and compare each option to find the correct answer?

---

> ### Author Response · Authors · 2024-11-21
> **The size of the synthetic data after filtering is kept consistent across different baselines and our method.**
>
> Reviewer: _”Can you compare the size of synthetic data you use for all methods? You mentioned that 32 reasoning paths are sampled for STaR, but how many of them are used for training after filtering?”_
>
> **Our response**:
>
> We thank the reviewer for this question since it points out that the size of the synthetic data after filtering is also a variable that could significantly affect model performance. We had this variable controlled in all the experiments of the current submission and we should have presented it in the experiment section. The table below lists the training data size that ReGenesis and STaR use for all training data sets:
>
> |      | GSM8K  | NumGLUE | ARC_c | ReClor | StrategyQA |
> |------|--------|---------|-------|--------|------------|
> | Size | 31,994 | 85,078  | 4,789 | 18,086 | 4,079      |
>
> The same training data size of ReGenesis and STaR indicates that the performance difference between ReGenesis and STaR is not caused by training data size difference.

---

> > ### Author Response · Authors · 2024-11-21
> > **ReGenesis Without Ground-Truth Answers: Effective and Outperforming the Baseline**
> >
> > Reviewer: _”ReGenesis still relies on access to a dataset of questions and even ground-truth answers, limiting its potential for fully autonomous reasoning and diminishing its broader impact.”_
> >
> > **Our response**:
> >
> > We agree that it can make regenesis less applicable if it has hard dependency on ground-truth answers. **Therefore, we have tested regenesis on the condition when ground-truth answers are absent and compared with existing baselines. Results are shown in Table 5. In this experiment setting, for both regenesis and baseline method, we use both self consistency and majority voting to replace ground truth answers for both methods. In all scenarios tested, regenesis outperformed the baseline.**
> >
> > We acknowledge that self-synthesizing reasoning prompts that can help model performance improvement,  is another important part of the challenge, which is the focus of our current on-going project. How to generate reasoning prompts to help model self-improve, is actually quite entangled with the reasoning path synthesis approach. We have found in our recent preliminary experiments that with regenesis as the reasoning path generation approach, “harder” prompts can help more compared to “more diverse” prompts, since regenesis guarantees generalization at the reasoning path generation step already. Then the problem pivots into how to self-generate hard prompts that the model has a chance to generate correct answers via majority voting and how to self-assess model failures beyond majority voting, which becomes a more independent scope that we are working on. We will share our observations with the community when we have a thorough conclusion.

---

> ### Author Response · Authors · 2024-11-21
> **[1/2] Investigating Why ReGenesis Significantly Outperforms STaR on GSM8k**
>
> Reviewer:  “_Why is there such a large difference between STaR and the proposed method on GSM8k? (46.3 vs 63.6)”_
>
> **Our Response**:
>
> We **first** conducted a **case study** focusing on questions where our method succeeded but STaR failed. Our analysis revealed that our method employs much  more diverse strategies, which are absent in LLMs trained with STaR.
>
> 1. ReGenesis answers questions by **systematically** breaking down the problem into smaller parts: gathering data, calculating potential increases, and comparing profits. This approach demonstrates critical thinking and analytical reasoning, effectively leading to the correct conclusion.
>   * **Example question**: A merchant wants to make a choice of purchase between 2 purchase plans: jewelry worth 5,000 or electronic gadgets worth 8,000. His financial advisor speculates that the jewelry market will go up 2.5% while the electronic gadgets market will rise 1.2% within the same month. If the merchant is looking to maximize profit at the end of this month by making a choice, how much profit would this be?
>   * **STaR answer**: Let's calculate the future value of both options: 1. If the merchant chooses jewelry worth 5,000, the future value will be: 5,000 + (0.025 * 5,000) = <<5000 + (0.025 * 5000)=525>>5,250.  2. If the merchant chooses electronic gadgets worth 8,000, the future value will be: 8,000 + (0.012 * 8,000) = <<8000 + (0.012 * 8000)=808>>8,080. Comparing both values, the merchant should choose electronic gadgets worth $8,000 since it has a higher future value of 8,080 compared to jewelry worth 5,250. The profit the merchant would make is 8,080 - 8,000 = <<8080-8000=80>>80. #### 80
>   * **ReGenesis Answer**: 1. Gather data: - Current prices: 5,000 (jewelry) and 8,000 (electronic gadgets) - Expected market growth rates: 2.5% for jewelry and 1.2% for electronic gadgets 2. Calculate the potential increase in value: - For jewelry: 2.5% of 5,000 = 125 - For electronic gadgets: 1.2% of 8,000 = 96 3. Calculate the potential profit for each option: - Profit for jewelry: 125 (increased value) - Profit for electronic gadgets: 96 (increased value) 4. Compare the potential profits: - The merchant will make a profit of 125 if they choose jewelry, or $96 if they choose electronic gadgets. 5. Evaluate other factors: - The merchant should consider additional factors such as personal preferences, budget, and other financial obligations when making the final decision. 6. Make a decision: - Based on the calculated profits, the merchant will make a higher profit by choosing jewelry. However, they should weigh this with their personal financial situation and other factors before making the final decision. The merchant will make a profit of 125 if they choose jewelry.
>
> 2. ReGenesis also uses more **logical reasoning processes**, which involves identifying a pattern.
>   * **Example question**: Alice likes to count the puffs of clouds in the sky while she eats her lunch outside at school. On Monday she counts just 3 puffs of clouds. Each day after that through Friday, though, she sees double the number of clouds in the sky as the day before. At the end of the week, how many clouds will she have counted in the sky at lunch across all five days?
>   * **STaR answer**: On Monday, Alice counts 3 puffs of clouds. Tuesday: 3 x 2 = <<3*2=6>>6 puffs. Wednesday: 6 x 2 = <<6*2=12>>12 puffs. Thursday: 12 x 2 = <<12*2=24>>24 puffs. Friday: 24 x 2 = <<24*2=48>>48 puffs. Adding all of them up: 3 + 6 + 12 + 24 + 48 = <<3+6+12+24+48=83>>83 puffs. #### 83
>   * **ReGenesis Answer**: 1. Identify the initial number of clouds Alice sees on Monday: 3 clouds. 2. Recall that each day after Monday, she sees double the number of clouds from the previous day. 3. Create a pattern for the number of clouds she sees on each day: Monday: 3 clouds; Tuesday: 3 * 2 = 6 clouds; Wednesday: 6 * 2 = 12 clouds; Thursday: 12 * 2 = 24 clouds; Friday: 24 * 2 = 48 clouds. 4. Add up the number of clouds she sees each day to find the total number of clouds across the week: Total clouds = Monday + Tuesday + Wednesday + Thursday + Friday; Total clouds = 3 + 6 + 12 + 24 + 48; Total clouds = 93 clouds. So, Alice will have counted 93 clouds in the sky at lunch across all five days.

---

> > ### Author Response · Authors · 2024-11-21
> > **[2/2] Investigating Why ReGenesis Significantly Outperforms STaR on GSM8k**
> >
> > 3. ReGenesis also conducts more **critical thinking** to make the answer correct.
> > * **Example Question**: Lee used to be able to run the 400-meter hurdles two seconds faster than Gerald would run the 400-meter hurdles.  But Gerald changed his diet, which improved his speed by 10%.  If Lee runs the 400-meter hurdles in 38 seconds, how fast can Gerald, with his improved diet, run the 400-meter hurdles, in seconds?
> > * **STaR answer**: Since Lee runs 2 seconds faster than Gerald, we can say that when they were running with their previous diets, Lee ran in t seconds and Gerald ran in t + 2 seconds, where t represents the time in seconds that Lee ran with his improved diet (which we don't know yet).
> > Their respective times in seconds with their old diets were related by: t + 2 = (t - 2) * 0.9 (Gerald's time increased by 10%). Solving for t: t = (2 / (1 - 0.9)) + 2 = 2 / 0.1 + 2 = <<2 / 0.1 + 2 = 20 + 2 = 22>>22.
> > Since Lee runs the 400-meter hurdles in 38 seconds with his improved diet, we know that: t = 38.
> > So, with his improved diet, Gerald can run the 400-meter hurdles in t + 2 = 38 + 2 = <<38+2=40>>40 seconds. #### 40
> > * **ReGenesis Answer**: 1. Establish the initial time difference between Lee and Gerald: - We know that Lee runs the 400-meter hurdles in 38 seconds, and Gerald used to be 2 seconds slower. - Therefore, Gerald’s initial time for the 400-meter hurdles was \(38 + 2 = 40\) seconds. 2. Account for the 10% improvement in Gerald’s speed: - A 10% improvement in speed means that Gerald's running time will decrease, as he will now run faster. - To calculate the new time, we need to find 90% of his original time (since he is now 10% faster, the remaining 90% represents his new time). - The formula for the new time is: \[ \text{New Time} = \text{Initial Time} \times 0.90 \] - Substituting the values: \[ \text{New Time} = 40 \times 0.90 = 36 \text{ seconds} \] Therefore, after improving his diet, Gerald can now run the 400-meter hurdles in 36 seconds, reflecting a 10% improvement in his speed.
> >
> > 4. ReGenesis also uses **reflective thinking** by ensuring the calculations align with the task requirement
> > * **Example Question**:Lee used to be able to run the 400-meter hurdles two seconds faster than Gerald would run the 400-meter hurdles.  But Gerald changed his diet, which improved his speed by 10%.  If Lee runs the 400-meter hurdles in 38 seconds, how fast can Gerald, with his improved diet, run the 400-meter hurdles, in seconds?
> > * **STaR answer**: Let's call the initial time that Gerald could run the 400m hurdles before changing his diet as t. Lee could run 2 seconds faster, so his time is t - 2 seconds. Since the difference between their times is 2 seconds, we can write the following equation: t - (t - 2) = 38. Solving this equation gives t = 38 + 2 = <<38+2=40>>40. Before changing his diet, Gerald ran the 400m hurdles in 40 seconds. Gerald improved his speed by 10%, so his new time is: 40 * (1 + 0.1) = 40 * 1.1 = <<40\*1.1=44>>44 seconds. #### 44
> > * **ReGenesis Answer**: To find out how fast Gerald can run the 400-meter hurdles after improving his diet by 10%, we can follow these steps: 1. Recall the initial difference in running times between Lee and Gerald: Lee runs 400-meter hurdles in 38 seconds, and Gerald is 2 seconds slower. So, Gerald's initial time was 38 + 2 = 40 seconds. 2. Calculate the improvement in Gerald's time due to his diet change: Since the improvement is a 10% increase in speed, we can find the new time by subtracting 10% of the initial time. The formula for this is: New Time = Initial Time - (Initial Time * Percentage Change). In this case, New Time = 40 - (40 * 0.10) = 40 - 4 = 36 seconds. So, with his improved diet, Gerald can run the 400-meter hurdles in approximately 36 seconds.
> >
> > **Finally**, we evaluated the accuracy of _Mistral-7B-Instruct-v0.3_ prompted with various reasoning guidelines on the GSM8K dataset. The results indicate that no single strategy is effective for all questions, necessitating the use of multiple strategies. As demonstrated, LLMs trained with ReGenesis show a broader range of answering strategies compared to those trained with STaR. We attribute this enhanced performance to the diverse answering strategies learned by the ReGenesis-trained LLM from its training data.
> > | Guidelines                 | GSM8K  |
> > |----------------------------|--------|
> > | How can I break down...    | 53.45% |
> > | Try creative thinking ...  | 20.30% |
> > | Critical Thinking...       | 47.56% |
> > | Use Reflective Thinking... | 42.69% |
> > | Use systems thinking...    | 39.44% |

---

> ### Author Response · Authors · 2024-11-24
> **Gentle reminder that the discussion period closes soon**
>
> This is a gentle reminder that the discussion period closes soon. We have thoroughly addressed the concerns raised, provided detailed rationales, and conducted additional experiments and case studies to strengthen our work. We would be greatly encouraged if the reviewers either raised their scores or engaged in further discussion with us. Thanks a lot.

---

> > ### Comment · Reviewer_w4BA · 2024-11-26
> >
> > Thanks for your response. The supplemented analysis on diversity has resolved my concerns and I have raised my score. I strongly recommend restructuring the paper to give more prominence to the analysis as a crucial part of your experiments, which may make your claim more valid and convincing.

---

> > > ### Author Response · Authors · 2024-11-27
> > >
> > > Thank you for your feedback and for raising your score. We're glad the additional analysis resolved your concerns. We will revise the paper in the camera-ready version to give more focus to this analysis, as you suggested. Your input is very helpful and much appreciated!

---

### Comment · Area_Chair_d6p6 · 2024-11-25

Dear Reviewers,

The rebuttal discussion period is coming to a close and the paper currently has a mix of positive and negative reviewers. The authors have spent a lot of time responding to each concern -- can you take a look at the author responses and let them know any remaining concerns you have?

Best,
AC

---

### Meta-Review · Area_Chair_d6p6 · 2024-12-23

**Metareview:**

This work proposes ReGenesis, an approach to enhance LLM reasoning abilities by self-synthesizing generalizable reasoning paths without human supervision or task-specific examples. Unlike prior methods that struggle with OOD generalization due to reliance on specific task examples, ReGenesis uses a progressive approach that builds from abstract reasoning guidelines to task-specific structures and paths. This approach encourages generalization across tasks, improving performance both in-domain and on OOD reasoning tasks. Extensive experiments/analysis show ReGenesis's adaptability across various LLMs, design configurations, and reasoning tasks especially in OOD settings. The reviewers have found the paper well-written, introducing a convincing approach for enhancing LLM reasoning performance and generality with ample empirical evidence. It's necessary for authors to include the additional analysis posted during rebuttal/discussion to make the approach clearer.

**Additional Comments On Reviewer Discussion:**

The discussion is effective as reviewers raised their scores based on the authors' responses. It's necessary for authors to include the additional analysis posted during rebuttal/discussion to make the approach clearer.

---

### Decision · Program_Chairs · 2025-01-22

Accept (Oral)